# Mind the Gap: Understanding the Modality Gap in Multi-modal Contrastive Representation Learning

**Weixin Liang**[*]
Stanford University
`wxliang@stanford.edu`

**Yuhui Zhang** [*]
Stanford University
`yuhuiz@stanford.edu`

**Yongchan Kwon** [*]
Columbia University
`yk3012@columbia.edu`

**Serena Yeung**
Stanford University
`syyeung@stanford.edu`

**James Zou**
Stanford University
`jamesz@stanford.edu`

## Abstract

We present *modality gap*, an intriguing geometric phenomenon of the representation space of multi-modal models. Specifically, we show that different data modalities (e.g. images and text) are embedded at arm's length in their shared representation in multi-modal models such as CLIP. Our systematic analysis demonstrates that this gap is caused by a combination of model initialization and contrastive learning optimization. In model initialization, we show empirically and theoretically that the representation of a common deep neural network is restricted to a narrow cone. As a consequence, in a multi-modal model with two encoders, the representations of the two modalities are clearly apart when the model is initialized. During optimization, contrastive learning keeps the different modalities separated by a certain distance, which is influenced by the temperature parameter in the loss function. Our experiments further demonstrate that varying the modality gap distance has a significant impact in improving the model's downstream zero-shot classification performance and fairness. Our code and data are available at https://modalitygap.readthedocs.io/

## 1 Introduction

Multi-modal models map inputs from different data modalities (e.g. image and text) into a shared representation space (Figure 1 (a)). It has garnered tremendous interest and excitement as a framework for data integration. As a prominent example pre-trained on a web-scale collection of images and natural language, OpenAI's CLIP model [39], has learned diverse visual concepts that can readily be transferred to downstream tasks through *prompting*: one can perform "zero-shot" visual classification by simply providing the names of the visual categories to be recognized.

In this work, we present the *modality gap* phenomenon: As shown in Figure 1 (b), CLIP's image embeddings and text embeddings are located in two completely separate regions of the embedding space. We find this phenomenon consistently across various multi-modal models, covering texts, natural images [39], videos [50], medical images [53], and amino-acid sequences [11]. Interestingly, this phenomenon still holds even when we embed using multi-modal models with *random* weights (Figure 1 (c)). While it might seem reasonable to attribute the gap to differences in data distributions or to the different encoder architectures, we showed that these factors are not the fundamental cause.

This paper provides a three-part explanation for the modality gap phenomenon. **(1)** The general inductive bias of deep neural architecture creates a *cone effect*: The effective embedding space is

---

[*]These three authors contributed equally.

36th Conference on Neural Information Processing Systems (NeurIPS 2022).

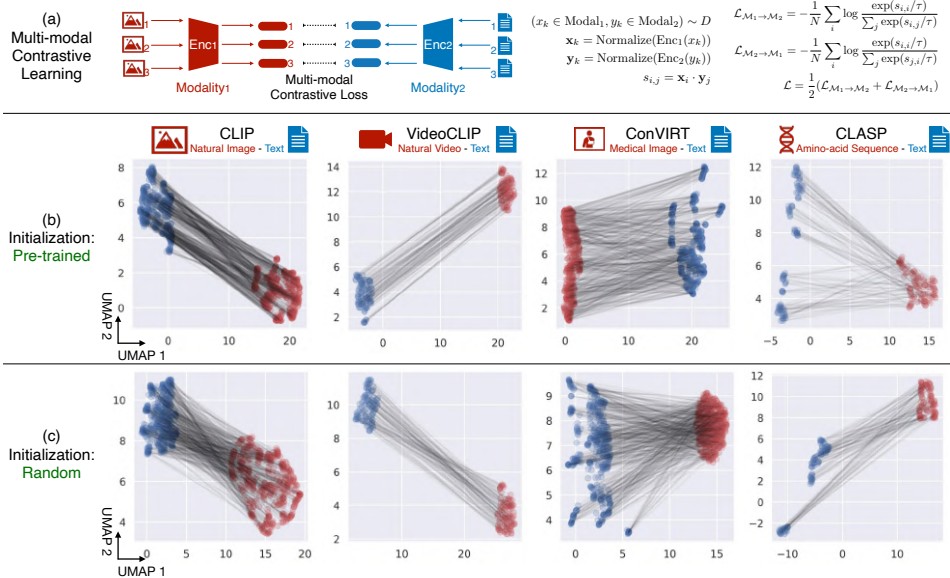

Figure 1: **The pervasive *modality gap* in multi-modal contrastive representation learning. (a) Overview of multi-modal contrastive learning.** Paired inputs from two modalities (e.g., image-caption) are sampled from the dataset and embedded into the hypersphere using two different encoders. The loss function is to maximize the cosine similarity between matched pairs given all the pairs within the same batch. **(b) UMAP visualization of generated embeddings from pre-trained models.** Paired inputs are fed into the pre-trained models and the embeddings are visualized in 2D using UMAP (lines indicate pairs). We observe a clear modality gap for various models trained on different modalities. **(c) UMAP visualization of generated embeddings from same architectures with random weights.** Modality gap exists in the initialization stage without any training.

restricted to a narrow cone for pre-trained models or models with random weights. **(2)** Different random initializations create different embedding cones. Since a multi-modal model consists of two encoders, which create different cones at random initialization, this explains how the modality gap is present at initialization. **(3)** The contrastive learning objective commonly used by multi-modal models preserves the gap. We support our explanations with theory and experiments. Our theoretical analysis shows that under mild assumptions, each neural network layer shrinks the angle between any pair of embedding vectors with high probability, thereby creating more narrow cones in deeper architectures. We further prove that different random initializations of model weights result in different cones. Interestingly, increasing the modality gap in models like CLIP can improve its downstream performance on several zero-shot learning and fairness tasks. The main objective of our paper is to i) empirically demonstrate the modality gap phenomenon across different data modalities and NN architectures; ii) explain how the gap arises and iii) show that the size of the gap can affect downstream applications. It is *not* our goal to propose a method to close the gap, since it's not clear that it's desirable to have no modality gap. Together, this paper makes the **following contributions**:

1. To the best of our knowledge, we demonstrate a general *modality gap* phenomenon for the first time. We show that this phenomenon holds across a wide spectrum of multi-modal models, covering texts, natural images, videos, medical images, and amino-acid sequences.
2. We demonstrate the significant implications of modifying the gap in downstream applications. By simply modifying the gap's distance, we can improve CLIP's zero-shot performance and fairness.
3. To explain modality gap, we provide a three-part explanation supported by extensive theoretical and empirical analyses. Our analyses also provide new insights on the cone effect, which we show is a general phenomenon for deep neural networks. Existing work focuses on *trained* language models and attributes the cone effect to the *optimization* under unbalanced word frequencies distribution. We demonstrate that this effect holds not only across various modalities and network architectures, but also on random noise inputs and random weights, which is not captured in previous work.

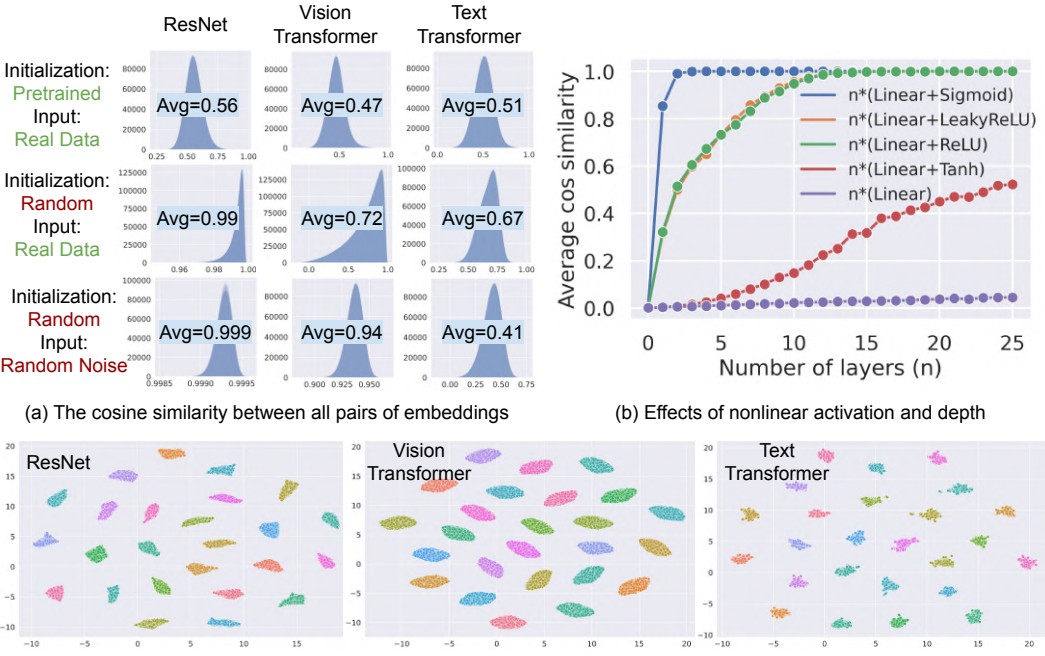

(a) The cosine similarity between all pairs of embeddings

(b) Effects of nonlinear activation and depth

(c) UMAP visualization of embeddings of 25 randomly initialized models on **real data** (color indicates random seed)

Figure 2: **The cone effect phenomenon. (a) Histograms of the cosine similarity between all pairs of embeddings across various settings.** The average cosine similarity is substantially larger than 0, indicating that the embedding space is a narrow cone. The cone effect also holds on randomly initialized models, and on random noise inputs. **(b) Effects of nonlinear activation and depth.** Inputs are 512-dim standard normal random vector. All MLP linear layers are $512 \times 512$, with both weight and bias randomly initialized from $\mathcal{N}(0, \frac{1}{512})$. Y axis is the average cosine similarity between pairs of embeddings. **(c) UMAP visualization of embeddings of 25 randomly initialized models (without training) on** *real* **data.** Each random initialization forms a distinctively different cone. *Real Data:* 5,000 image-caption pairs from the validation set of MSCOCO Caption. *Random Noise:* Gaussian noise from the standard normal distribution as images, uniformly random integer sequences as texts.

4. We mathematically characterize the contraction mapping induced by linear layers with ReLU non-linearities to explain the cone effect. Our theory matches well with experiments and provides insights for understanding the general inductive biases of deep neural networks.

## 2 The Cone Effect Induces A Modality Gap

### 2.1 The Narrow Cone of Embeddings

In order for modality gap to exist, the embeddings from a encoder should be concentrated around a subregion of the full embedding space—otherwise, the embeddings from different encoders would overlap. Motivated by this, we begin our investigation by showing that the modality gap already arises at random model initialization due to the *cone effect*: The effective embedding space is restricted to a narrow cone for trained models and models with random weights. To demonstrate this, we extract 5,000 embeddings from the final layer of 3 pre-trained models respectively (ResNet, Vision Transformer, Text Transformer)[2] on MSCOCO Caption [8]. We then compute the cosine similarity between all possible pairs of the 5,000 embeddings within each model (Figure 2 (a)). We found that both the average cosine similarity (0.56, 0.47, 0.51 respectively for the 3 models) and the minimum cosine similarity (0.23, 0.05, 0.01) are positive. These results indicate that the embedding space is a narrow cone.

---

[2]ResNet embeddings are extracted before the final linear layer. We use ResNet-18 pre-trained on ImageNet, Vision Transformer and Text Transformer from pre-trained CLIP

In the literature, the cone effect has been observed in the language representations from language models (e.g., BERT) [12]. A common explanation is that the *unbalanced* distribution of word frequencies biased the *optimization* [15, 33]. However, we found that the cone effect still exists in models with random weights (Figure 2 (c)). In fact, the average cosine similarity there is even *higher* than in trained models. For example, any two embeddings from a randomly initialized ResNet have on average an almost perfect (0.99) cosine similarity. Interestingly, the cone effect still holds when the input data is random noise[3], indicating that unbalanced data distribution suggested in previous works is not necessary for the cone effect. Together these experiments suggest that the cone effect reflects a more general inductive bias of deep networks than might be previously appreciated.

**How narrow is the cone in 512-dim representation space?** We clarify that a cosine similarity with $0.56$ already indicates that the embedding space is actually an extremely narrow cone in the 512-dimensional feature space. Consider the fraction of surface area in a unit hypersphere: In 2D, arccos(0.56)=55.94°, indicating that a cosine similarity of 0.56 can "occupy" 55.94°/360°=15.53% of the 2D unit circle. In 3D, a cosine similarity of 0.56 can "occupy" $\frac{2\pi r^2(1-\cos\frac{55.94°}{2})}{4\pi r^2}$=3.34% of the 3D unit sphere. In 512D, a cosine similarity of 0.56 can "occupy" less than $\frac{1}{2^{512}}$ fraction of the surface area in a unit 512D hypersphere. These evidences show that the effective embedding space is restricted to an extremely narrow cone.

## 2.2 The effects of non-linear activation on cone effect

**Design** To study the effects of non-linear activation functions on the cone effect, we randomly initialized various MLPs with different non-linearities or without non-linearities. The inputs of the MLPs are 512-dim standard normal random vectors. All MLP linear layers are $512 \times 512$, with both weight and bias randomly initialized from $\mathcal{N}(0, \frac{1}{512})$, here we denote a Gaussian distribution with mean $\mu$ and variance $\sigma^2$ by $\mathcal{N}(\mu, \sigma^2)$.

**Results** As shown in Figure 2 (b), MLPs without non-linear activation shows little cone effect. However, with non-linearity, the average cosine similarity increases *rapidly* as the number of layers increases. For example, the average cosine similarity reaches $0.99$ for a 2-layer MLP with Sigmoid. These results indicate that the non-linear activation functions play a crucial role in the cone effect.

Although it is easy to see that ReLU makes every coordinate non-negative, and thus cosine similarity after ReLU is guaranteed to be non-negative, we highlight that none of the 3 models in Figure 2 (a) has ReLU as the final layer before embedding extraction[4]. In addition, although all 3 models incorporate normalization layers such as batch norm [23] and layer norm [4] in their architectures, we still observe the cone effect. Further analyzing the connection between normalization and the cone effect is an interesting direction of future work.

## 2.3 Different random initializations create different cones

Next, we study the effect of different random initialization on the cone effect. In Figure 2 (c), we randomly initialized a model 25 times, and plotted its extracted embeddings on the same *real data* (i.e., MSCOCO Caption) via UMAP visualization [41]. We found that each random initialization forms a distinctively different cone. This phenomenon holds across various neural network architectures and input modalities (ResNet, Vision Transformer or Text Transformer), on ImageNet-pretrained models (Supp. Figure 13), on PCA visualization (Supp. Figure 7), or with random noise inputs (Supp. Figure 5). Since a multi-modal model consists of two encoders, which creates different cones at random initialization, this explains how the modality gap is present at initialization. *While it might seem reasonable to attribute the modality gap to differences in data modalities [21], Figure 2 (c) shows the gap still exists even if the two encoders operate on the exact same data in the exact same modality. Therefore, the gap can exist without different modalities, and we emphasize that the modality gap phenomenon is non-trivial to understand.*

---

[3]Standard normal distribution for vision models, and uniformly random integer sequences for text models.

[4]The last 3 layers are Conv2d, BatchNorm2d, AdaptiveAvgPool2d for ResNet-18 (not counting last fc); Linear, LayerNorm, LayerNorm for Vision Transformer in CLIP; QuickGELU, Linear, LayerNorm for Text Transformer in CLIP.

## 3   Theoretical analysis

Here, we theoretically investigate the cone effect phenomenon. We show that (i) the cosine similarity increases as the layer gets deeper and (ii) the variance of an intermediate output mostly come from the model's random initialization.

We first define some notations. We denote the ReLU activation by $\phi(x) := \max(x, 0)$ for $x \in \mathbb{R}$, and we extend it by considering element-wise operation $\phi(\mathbf{x}) := (\phi(x_1), \ldots, \phi(x_k))^T = (\max(x_1, 0), \ldots, \max(x_k, 0))^T$ for a multivariate input $\mathbf{x} = (x_1, \ldots, x_k)^T \in \mathbb{R}^k$ and $k \in \mathbb{N}$. The cosine similarity between two vectors $u, v \in \mathbb{R}^k$ is defined as $\cos(u, v) := \frac{u^T v}{\|u\|\|v\|}$ where $\|u\| = (u^T u)^{1/2}$. Lastly, we set $[k] := \{1, \ldots, k\}$ for $k \in \mathbb{N}$.

**Each network layer increases cosine similarity.**   We study how the cosine similarity between two intermediate layer outputs changes when weight and bias terms in an MLP are fixed. The following theorem shows that with a high probability cosine similarity increases after one feedforward computation when the number of nodes in the output layer is large.

**Theorem 1** (Monotonicity of cosine similarity). *Suppose $u, v \in \mathbb{R}^{d_{\text{in}}}$ are any two fixed vectors such that $\|u\| = r\|v\|$ for some $r > 0$, $\mathbf{W} \in \mathbb{R}^{d_{\text{out}} \times d_{\text{in}}}$ is a random weight matrix where each element $\mathbf{W}_{k,l} \sim \mathcal{N}(0, d_{\text{out}}^{-1})$ for $k \in [d_{\text{out}}]$, $l \in [d_{\text{in}}]$, and $\mathbf{b} \in \mathbb{R}^{d_{\text{out}}}$ is a random bias vector such that $\mathbf{b}_k \sim \mathcal{N}(0, d_{\text{out}}^{-1})$ for $k \in [d_{\text{out}}]$. If $\cos(u, v) < \left(\frac{1}{2}\left(r + \frac{1}{r}\right)\right)^{-1}$, then the following holds with probability at least $1 - O(1/d_{\text{out}})$.*

$$\cos(\phi(\mathbf{W}u + \mathbf{b}), \phi(\mathbf{W}v + \mathbf{b})) > \cos(u, v).$$

Theorem 1 shows that the cosine similarity between two vectors increases with a high probability after one feedforward computation consisting of a linear transformation and ReLU computation. This matches well with the result in Figure 2 (b) where the cosine similarity between samples increases as the intermediate layer gets farther from the input.

The bound condition on $\cos(u, v)$ in Theorem 1 asks that the two vectors before the layer computation are not too close to each other in terms of the direction. This is because the random bias addition can slightly change the angle between the two vectors, leading to a small decrease in cosine similarity when the previous layer's cosine similarity is too high. This condition is plausible in practice because the $\ell^2$-norm of intermediate layer outputs is close to one with a high probability when the $\ell^2$-norm of input data is one [1, Lemma 7.1]. Given that the norm ratio $r$ is close to one, the upper bound condition for $\cos(u, v)$ is likely to hold because $(\frac{1}{2}(r + \frac{1}{r}))^{-1}$ is close to 1.

**Effect of random initialization**   We now examine the variance of an intermediate output and explain that the variance is mainly due to random initializations as in Figure 2 (c). To be more specific, we denote an intermediate layer output by $h_\Theta(U) \in \mathbb{R}$ for some input datum $U$. Here, $\Theta$ denotes all the random weights and biases that are used in $h_\Theta(U)$. The variance of $h_\Theta(U)$ can be decomposed as

$$\text{Var}[h_\Theta(U)] = \underbrace{\mathbb{E}[\text{Var}[h_\Theta(U) \mid \Theta]]}_{\text{Due to the randomness of data}} + \underbrace{\text{Var}[\mathbb{E}[h_\Theta(U) \mid \Theta]]}_{\text{Due to random initializations}}.$$

Here, the inner and outer expectations are over the data $U$ and the random weights $\Theta$, respectively. The first term on the right hand side explains the within variance after fixing one random initialization, quantifying the randomness of data. In contrast, the second term explains the variance due to different random initializations. The following theorem considers the ratio of the second term to the total variance and shows that the ratio can be very close to one when a deep neural network model is used.

**Theorem 2** (Informal; Variance due to different random initializations). *Let $h_\Theta(U)$ be an intermediate layer output with an input data $U$ with $\|U\| = 1$. Under mild assumptions on $\Theta$, the set of all the random weights and biases, the following inequality holds.*

$$\frac{\text{Var}[\mathbb{E}[h_\Theta(U) \mid \Theta]]}{\text{Var}[h_\Theta(U)]} \geq \beta,$$

*where $\beta$ is a constant that captures the average cosine similarity of previous layer outputs.*

Theorem 2 shows that the ratio of the variance due to different random initializations to the total variance is bounded below by the average cosine similarity of previous layer outputs. As Figure 2 (b) illustrated, the average cosine similarity of an intermediate layer output often approaches to one as the layer gets deeper. Accordingly, the lower bound $\beta$, which captures the average cosine similarity, is close to one when a neural network is deep enough. In Appendix D, we elaborate on the relationship between $\beta$ and the cosine similarity, and provide a detailed statement of the Theorem.

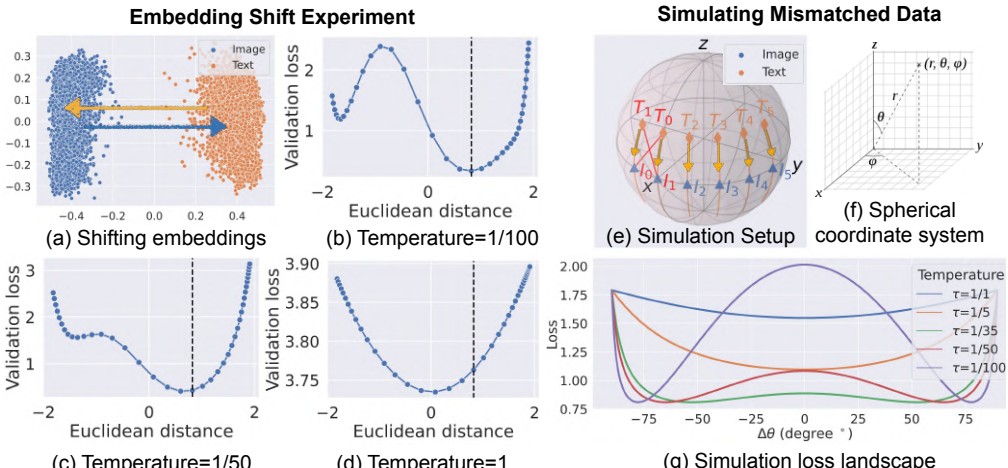

Figure 3: **Contrastive learning preserves modality gap. (a) Embedding shift experiment.** To probe the loss landscape of CLIP, we manually shift the image embeddings and text embeddings towards closing the gap. **(b-d) The loss landscapes under different temperatures.** Y axis indicates the contrastive loss. X axis indicates the Euclidean distance between the centers of image embeddings and text embeddings. The vertical dash line $x = 0.82$ indicates CLIP's original distance between image and text embeddings (i.e., without any shifting). Note that in CLIP, the image embeddings and text embeddings are L2-normalized (Supplementary Figure 12). In other words, the image and text embeddings of CLIP are always on the unit sphere. **(e-g) Simulation analysis for the loss landscape.** Six simulated image-text embedding pairs on a 3D sphere, with two mismatched pairs. Text embeddings are shifted towards closing the modality gap (i.e., modifying $\theta$).

# 4 Contrastive learning preserves modality gap

## 4.1 Background: Contrastive Loss

Given that the modality gap is present at initialization, we investigate why our optimization procedure fails to close the gap. We begin by reviewing contrastive learning, which is a commonly used training strategy for multi-modal models [53, 50, 34]. We illustrate with CLIP due to its wide usage.

Given a batch of $N$ (image, text) pairs, CLIP learns to predict which of the $N \times N$ possible (image, text) pairs are aligned. In other words, CLIP learns to maximize the cosine similarity of the image and text embeddings of the $N$ real pairs in the batch while minimizing the cosine similarity of the embeddings of the $N^2 - N$ incorrect pairs. Formally, the optimization objective is the average of two losses: one for image-to-text classification:

$$\mathcal{L}_{\mathcal{I} \to \mathcal{T}} = -\frac{1}{N} \sum_{i=1}^{N} \log \frac{\exp(\mathbf{x}_i \cdot \mathbf{y}_i / \tau)}{\sum_{j=1}^{N} \exp(\mathbf{x}_i \cdot \mathbf{y}_j / \tau)}$$

and the other for text-to-image classification:

$$\mathcal{L}_{\mathcal{T} \to \mathcal{I}} = -\frac{1}{N} \sum_{i=1}^{N} \log \frac{\exp(\mathbf{x}_i \cdot \mathbf{y}_i / \tau)}{\sum_{j=1}^{N} \exp(\mathbf{x}_j \cdot \mathbf{y}_i / \tau)}$$

Here, $\mathbf{x}_i$ and $\mathbf{y}_j$ are the L2-normalized embedding of image in the $i$-th pair and that of text in the $j$-th pair, respectively. $\tau$ is a learned temperature parameter to scale the logits. The final learned temperature is $\tau = \frac{1}{100}$ in CLIP. See additional illustration in Figure 1(a) and Supp. Figure 12.

## 4.2 Embedding Shift Experiment

**Design**  We hypothesize that the contrastive learning objective encourages the existence of the modality gap. To testify this hypothesis, we design a loss landscape probing experiment on $n = 5,000$ image-caption pairs[5] from the validation set of MSCOCO Caption dataset. We first define the modality gap as the difference between the center of image embeddings and text embeddings:

$$\vec{\Delta}_{\text{gap}} = \frac{1}{n} \sum_{i=1}^{n} \mathbf{x}_i - \frac{1}{n} \sum_{i=1}^{n} \mathbf{y}_i$$

where $\mathbf{x}_i$ and $\mathbf{y}_i$ are the L2-normalized image embedding and text embedding. We then manually shift every text embedding and image embedding towards closing the modality gap (Figure 3 (a)). After shifting, we re-normalize each embedding to the unit hypersphere:

$$\mathbf{x}_i^{\text{shift}} = \text{Normalize}(\mathbf{x}_i - \lambda \vec{\Delta}_{\text{gap}}), \quad \mathbf{y}_i^{\text{shift}} = \text{Normalize}(\mathbf{y}_i + \lambda \vec{\Delta}_{\text{gap}}).$$

We vary the scalar $\lambda$ to produce different amounts of shifts. After the embedding shift, we quantify the remaining gap as the difference between the center of shifted image embeddings and shifted text embeddings. The gap distance before shifting is $\|\vec{\Delta}_{\text{gap}}\| = 0.82$. Here Euclidean distance is a intuitive metric because in CLIP, the image embeddings and text embeddings are L2-normalized (Supplementary Figure 12). In other words, the image and text embeddings of CLIP are always on the unit sphere. Specifically, for any $n$-dimensional vectors $x$ and $y$, the cosine similarity is given as $\cos(x, y) = x^T y$, and the Euclidean distance is given as $(x - y)^T (x - y) = 2(1 - x^T y)$. Therefore, they have a functional relationship as $\text{Euclideandistance}(x, y) = 2(1 - \cos(x, y))$. When the angle between $x$ and $y$ is less than $\pi/2$, which is the case as embeddings are in a narrow cone, the small Euclidean distance directly means a high cosine similarity.

**Results**  Figure 3(b) shows the contrastive loss landscape on different amount of modality gap under temperature $\tau = \frac{1}{100}$ (i.e., CLIP's learned final temperature). We found that the default gap distance $\|\vec{\Delta}_{\text{gap}}\| = 0.82$ actually achieves the global minimum, and shifting toward closing the gap *increases* the contrastive loss. Interestingly, there is a local minimum when we shift the text embeddings to the opposite side in a "back-to-back position." Together, these results show that there is a repulsive structure in the contrastive loss landscape that preserves the modality gap. However, when the temperature increases (Figure 3(c,d)), the repulsive structure and the local minimum gradually disappear, and closing the gap becomes more optimal. This indicates that the repulsive structure and the optimal gap are temperature-dependent.

**Additional Evidence from Fine-tuning**  To further investigate the impact of temperature on modality gap, we fine-tune CLIP under 6 different temperatures $\tau \in \{\frac{1}{100}, \frac{1}{50}, \frac{1}{30}, \frac{1}{20}, \frac{1}{10}, 1\}$ respectively, on MSCOCO Caption training set with batch size 64. We found that a high temperature ($\tau \in \{\frac{1}{10}, 1\}$) in fine-tuning significantly reduces or closes the gap, while a low temperature does not. The gap distance $\|\vec{\Delta}_{\text{gap}}\|$ decreases monotonically with increasing temperature (Supp. Figure 8).

## 4.3 Simulating mismatched data

**Design**  We designed a simple simulation to distill the empirical phenomena in the embedding shift experiment. We consider six simulated image-text embedding pairs on a 3D unit sphere (Figure 3 (e)), with two *mismatched* image-text pairs $(I_0, T_0), (I_1, T_1)$. Here "mismatched" means correct pairs are $(I_0, T_0)$ and $(I_1, T_1)$ but $I_0$ is closer to $T_1$ and $I_1$ is closer to $T_0$. We fix the image embeddings while shifting the text embeddings downwards to close the gap (i.e., modifying $\theta$, see more details in Appendix A).

**Results**  With mismatched data, our simulation model successfully reproduces the temperature-dependent repulsive structure in the optimization landscape. When we remove the mismatch, the repulsive structure disappears (Supp. Figure 9). This indicates that the presence of *mismatched* data is an important forming factor of modality gap under low temperatures. Although the mismatch here is simulated, in practice mismatched data are common (e.g., hard-to-differentiate images/captions or annotation errors). Investigating how and to what extent the multimodal data misalignment could

---

[5]Here we evaluated CLIP with batch size 50.

| Dataset | Original gap | Modified gap | Direction |
|---|---|---|---|
| **Coarse-grained Classification** | | | |
| CIFAR10 | 0.9013 | **0.9081** | ↑ |
| CIFAR100 | 0.6658 | **0.6737** | ↓ |
| **Fine-grained Classification** | | | |
| EuroSAT | 0.5410 | **0.5645** | ↓ |
| **Optical Character Recognition** | | | |
| SVHN | 0.5389 | **0.5396** | ↑ |
| HatefulMemes | 0.5800 | **0.5811** | ↑ |

Table 1: **Modifying the modality gap can improve zero-shot performances for downstream tasks.** Number indicates top-1 accuracy. Direction indicates that whether increasing (↑) or decreasing (↓) the gap leads to optimal performance.

| Denigration Biases | Original gap | | | Modified gap | | |
|---|---|---|---|---|---|---|
| | Crime related | Non human | Sum | Crime related | Non human | Sum |
| Black | 1.0% | 0.1% | **1.1%** | 0.8% | 0.1% | **1.0%** |
| White | 15.5% | 0.2% | **15.7%** | 13.2% | 0.4% | **13.7%** |
| Indian | 1.2% | 0.0% | **1.2%** | 1.1% | 0.0% | **1.1%** |
| Latino | 2.8% | 0.1% | **2.8%** | 1.9% | 0.1% | **2.0%** |
| Middle Eastern | 6.3% | 0.0% | **6.3%** | 5.2% | 0.0% | **5.2%** |
| Southeast Asian | 0.5% | 0.0% | **0.5%** | 0.3% | 0.0% | **0.3%** |
| East Asian | 0.7% | 0.0% | **0.7%** | 0.6% | 0.0% | **0.6%** |

Table 2: **Modifying the modality gap reduces biases for all races.** Number indicates the fraction FairFace images whose top-1 prediction is offensive. Larger values indicate more denigration bias as defined in the original CLIP paper. Increasing the gap from $0.82$ to $0.97$ reduces denigration harms consistently for all races.

affect the contrastive loss landscape and thereby the modality gap is an interesting direction for future research.

## 4.4 Initialization vs Optimization

**Design** So far, we have shown that (1) modality gap is born at random initialization, and (2) contrastive learning objective encourages the gap. To explore how the final modality gap is affected by a combination of both factors, we train two CLIP models from scratch: one model uses random initialization, where the gap is large $\|\vec{\Delta}_{\text{gap}}\| = 1.1891 \pm 0.0017$ because of the cone effect discuss in Sec. 2; another model amends the gap at the initialization by transforming text embeddings to be close to the image embeddings, where the gap is almost zero $\|\vec{\Delta}_{\text{gap}}\| = 0.0388 \pm 0.0351$. Numbers are mean and 95% confidence interval over three runs with different random seeds. The transformation we applied is a common method to align multilingual word embeddings [31]. More specifically, given image embedding $\mathbf{x}$ and text embedding $\mathbf{y}$, we apply an orthogonal matrix to text embedding $\mathbf{y}' = W\mathbf{y}$ and compute the multi-modal contrastive loss on $\mathbf{x}$ and $\mathbf{y}'$. The orthogonal matrix minimizes the distance between image embeddings and transformed text embeddings: $W = \arg\min_{W \in O_D} \|X - YW\|$ where $X, Y \in \mathbb{R}^{N \times D}$ are image embeddings and text embeddings generated from $N$ image-caption pairs, and $O_D$ is the set of $D$-dimensional orthogonal matrix.

**Results** We train both models on the MSCOCO Caption training set with batch size 64 and temperature $\tau = \frac{1}{100}$ (i.e., CLIP's learned temperature). After training, the original model gap changes from $1.1891 \pm 0.0017$ to $1.2991 \pm 0.0389$, while the amended model gap changes from $0.0388 \pm 0.0351$ to $0.7457 \pm 0.0633$. Numbers are 95% confidence interval over three runs with different random seeds. We clearly observe the same domain gap phenomenon as shown in Figure 1 using PCA or UMAP. This experiment shows that the final domain gap is caused by both initialization and optimization. When we ablate the domain gap at the initialization, the loss will still encourage the gap, but the gap distance is only 57% compared to the model without amending the gap.

## 5 Modality Gap Implications

### 5.1 Zero-shot performance

**Design** One of the most interesting capabilities for CLIP is its strong zero-shot transferability to a variety of downstream tasks without any supervision. We study whether changing the gap will affect CLIP (ViT-B/16)'s performances on various downstream tasks, including coarse-grained classification (CIFAR10 and CIFAR100), fine-grained classification (EuroSAT [22]), and optical character recognition (SVHN, HatefulMemes [28]). Metric and prompt for each task are shown in Supp. Table 3. Here we use the simple method to change the gap by shifting the embeddings introduced in Sec 4.2. The main objective of our paper is to understand the modality gap phenomenon, a general inductive bias that holds across various data modalities and NN architectures. The goal of our paper is *not* to propose a method to close the gap and to improve downstream performance.

**Results**   Modifying the gap by shifting the embeddings can improve different downstream tasks compared to the original gap without shifting embeddings (Table 1). Details of performance vs gap distance curves are shown in Supp. Figure 10. We leave more methods to change the gap and more analysis of the relation between gap distance and downstream task performance to future work.

## 5.2   Fairness

**Design**   We follow the bias evaluation setup in the CLIP paper to evaluate denigration harms [39, Sec. 7.1]. We performed zero-shot evaluations on CLIP (ViT-B/32) on the evaluation set of the Fair-Face dataset [26], which has 10,954 images. In addition to the 14 FairFace classes (e.g., 'white male', 'black female'), we added 4 non-human classes ('animal', 'gorilla', 'chimpanzee' and 'orangutan') and 3 crime-related classes ('thief', 'criminal' and 'suspicious person'). The text prompts are attached in Appendix (Supp. Figure 11). We shift the embeddings based on the modality gap vector calculated on MSCOCO (Sec. 4.2). We report the fraction FairFace images whose top-1 prediction is offensive.

**Results**   We found that increasing the gap from $0.82$ to $0.97$ *reduces* denigration harms consistently for *all* races (Table 5). Meanwhile, we only observe a minor $0.0008$ top-1 accuracy drop (Appendix B.2). It is encouraging that a simple gap offsetting approach can lead to a consistent bias reduction across all races on such a complex model (i.e., CLIP)[6]. Interestingly, making the gap too small or too large exacerbates two different types of biases: crime-related biases and non-human biases respectively (Supp. Table 4).

## 6   Related Work

**Contrastive Representation Learning**   Contrastive representation learning learns an embedding space where similar objects are closer than dissimilar ones, and has achieved great success in vision [7, 20, 6, 9], language [40, 16], and graph [51, 38]. However, as contrastive learning is still an emerging representation learning technique, we still lack comprehensive theoretical and empirical understandings about why contrastive learning works. [48] proposed two ideal objectives for contrastive representation space: alignment (similar samples have similar features) and uniformity (features are uniformly distributed on the hypersphere), and demonstrated these two objectives are highly correlated with downstream task performances. [46] show that low temperatures increase the model's penalty on hard negative examples, and thus increase uniformity and decrease tolerance (the closeness of semantically similar samples). These analyses mostly focus on unsupervised contrastive learning on a single modality. Orthogonal to their work, we show that multi-modal contrastive learning with low temperatures and mismatched data encourages the modality gap.

**Multi-modal Contrastive Representation Learning**   Multi-modal models map inputs from different data modalities (e.g. image and text) into a shared representation space [53, 50, 34, 24, 11]. It has garnered tremendous interest and excitement as a framework for data integration. These models are often pre-trained with contrastive loss [45], as [39] showed that the contrastive learning is $12\times$ more efficient than the generative approaches. We demonstrate an intriguing geometric phenomenon of the representation space of these multi-modal models, and provide a three-part explanation supported by theory and experiments. The idea of mapping images and text into a shared embedding space has been explored in earlier works [42, 49]. There have been recent efforts in formulating images and text embeddings as metric learning [14], multilabel classification [25], n-gram language learning [32], and captioning [10]. Recently there has there has also been work in using a unified encoder to fuse different data modalities [19]. Research into how the modality gap phenomenon generalizes to the multi-modal representations obtained by these alternative methods, or even uni-modal settings with teacher and student model [44, 5] would be a promising direction for future work.

**Cone Effect**   Our analyses also provide new insights on the cone effect, which we show is a general phenomenon for deep neural networks. Existing work focuses on the language representations of *trained* language models such as BERT and GPT-2 [12, 15, 33]. Given that isotropy has both theoretical and empirical benefits for static embeddings [35], the extent of anisotropy in contextualized

---

[6][39] evaluated a private version of CLIP, and thus their numbers deviate from ours. This is a known issue in the community: https://github.com/openai/CLIP/issues/157

representations is surprising [12]. It has been shown that the cone effect limits the expressiveness of the language representations. Post-processing methods [33, 43, 2, 35] or modified training objective [15, 47, 16] alleviate the cone effect and improve downstream performance. Existing work attributes the cone effect to the *optimization* under unbalanced word frequencies distribution [15, 33]. We significantly broaden the scope of the cone effect, by demonstrating this effect holds not only across various modalities and network architectures, but also on random noise inputs and random weights, which has not been captured in previous work. We mathematically characterize the contraction mapping induced by linear layers with ReLU non-linearities to explain the cone effect. Our theory matches well with experiments and provides insights for understanding the general inductive biases of deep neural networks.

# 7 Discussion

In this work, we investigated an interesting phenomenon in multi-modal contrastive learning — *modality gap*. We analyzed why the gap exists, i.e., the joint effect of model initialization and optimization, and why studying the gap is important, i.e., it can affect the downstream task performance and fairness. Our work raises several basic questions about representation learning, contrastive learning, and multi-modal contrastive representation learning. For representation learning, prior research in NLP has shown that alleviating the cone effect improves downstream performance. As our work significantly broadens the scope of the cone effect, methods for alleviating the cone effect in other modalities to improve ML performance is an interesting direction of future research.

For contrastive learning, our embedding shifting, simulation, and fine-tuning experiments all show that the contrast loss landscape is heavily influenced by temperature. Recent work has found that temperature directly controls the uniformity and affinity of the uni-modal representation space [46]. Our study provides a complementary understanding of the multi-modal representation space. Development of geometric methods for evaluation of representations [37, 30] to further capture the geometric landscape of the modality gap is an interesting direction of future work.

For multi-modal contrastive representational learning, we find that changing the modal gap can affect performance and fairness on downstream tasks. Interestingly, having *larger gap* can help some fairness and zero-shot learning applications. The main objective of our paper is to demonstrate the modality gap phenomenon and explain contraction mapping contribute to this. Systematic analysis of the impact of the gap on applications is an important direction of future work.

## Reproducibility Statement

We provide open-source implementation of our work at `https://github.com/Weixin-Liang/Modality-Gap`. The implementations will enable researchers to reproduce the modality gap described here as well as run their own analyses on additional cross-modal models. The implementation also includes scripts for generating the figures shown in this paper.

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
