# A    Contrastive learning preserves modality gap

## A.1    Simulating Mismatched Data

In Sec. 4.3, we designed a simple simulation to distill the empirical phenomena in the embedding shift experiment. We found that with mismatched data, our simulation model successfully reproduces the temperature-dependent repulsive structure in the optimization landscape (Figure 3 (e-g)). Here we present another simulation where we remove the mismatch (Supp. Figure 9). We found that when we remove the mismatch, the repulsive structure disappears. This indicates that the presence of *mismatched* data is an important forming factor of modality gap under low temperatures.

For both Figure 3 (e-g) and Supp. Figure 9, all embeddings are on the 3D unit sphere (i.e., $r = 1$). The spacing between adjacent image-text pairs is $\Delta\phi = 15°$. All image vectors are fixed, and located on the equator (i.e., $\theta = 90°$). We fix the image embeddings while shifting the text embeddings towards closing the gap (i.e., modifying $\theta$). Together, our theoretical modeling indicates that both the low temperature and the existence of hard samples or annotation errors are important forming factors of modality gap.

# B    Modality Gap Implications

## B.1    Zero-shot Performance

In Sec. 5.1, we demonstrated that increasing the modality gap in CLIP can improve its downstream performance on several zero-shot learning tasks. The downstream tasks we evaluated include coarse-grained classification (CIFAR10 and CIFAR100), fine-grained classification (EuroSAT [22]), and optical character recognition (SVHN, HatefulMemes [28]). Metric and prompt for each task are shown in Appendix Table 3. Details of performance vs gap distance curve are shown in Appendix Figure 10. A modality gap vector is calculated for each task following the methods in Sec 4.2.

## B.2    Fairness

In Sec. 5.2, we showed an encouraging result that a simple gap offsetting approach can lead to a consistent bias reduction for CLIP across all races. Meanwhile, we only observe a minor $0.0008$ top-1 accuracy drop, from $0.5817$ to $0.5739$. We show text prompts we used in Supp. Figure 11. Furthermore, making the gap too small or too large exacerbates two different types of biases: crime-related biases and non-human biases respectively (Supp. Table 4). Making the gap too small ($d = 0.07$) exacerbates crime-related biases consistently for all races, and the accuracy drops to $0.5599$. Making the gap too large ($d = 1.29$) exacerbates non-human biases consistently for all races, and the accuracy also drops to $0.4083$.

# C    The bigger picture: Why studying the modality gap is important

There has been tremendous recent interest and excitement in studying the inductive bias of neural networks mathematically and empirically [13]. For example, an influential line of research shows that neural networks can easily fit random labels [52], and SGD provides an inductive bias of "implicit regularization" by favoring minima that is flatter [27] and closer to the initialization [36]. Another impactful line of research shows that neural networks trained on natural scenes are biased towards texture [18], and exhibit gestalt closure similar to human perception, which is an inductive bias long-studied in the Psychology literature [29]. Researchers have also shown that neural networks favor "shortcut learning", which may be a common characteristic of learning systems, biological and artificial alike, as known in Comparative Psychology, Education and Linguistics [17, 3]. Our paper is positioned to be part of this broad and exciting trend of studying the inductive bias of neural networks by analyzing the modality gap phenomenon which occurs consistently in multi-modal contrastive representation learning.

$$(x_k \in \text{Domain}_1, y_k \in \text{Domain}_2) \sim \mathcal{D}$$

$$\mathbf{x}_k = \text{Normalize}(\text{Enc}_1(x_k))$$

$$\mathbf{y}_k = \text{Normalize}(\text{Enc}_2(y_k))$$

$$s_{i,j} = \mathbf{x}_i \cdot \mathbf{y}_j$$

$$\mathcal{L}_{\mathcal{D}_1 \to \mathcal{D}_2} = -\frac{1}{N}\sum_i \log\frac{\exp(s_{i,i}/\tau)}{\sum_j \exp(s_{i,j}/\tau)}$$

$$\mathcal{L}_{\mathcal{D}_2 \to \mathcal{D}_1} = -\frac{1}{N}\sum_i \log\frac{\exp(s_{i,i}/\tau)}{\sum_j \exp(s_{j,i}/\tau)}$$

$$\mathcal{L} = \frac{1}{2}(\mathcal{L}_{\mathcal{D}_1 \to \mathcal{D}_2} + \mathcal{L}_{\mathcal{D}_2 \to \mathcal{D}_1})$$

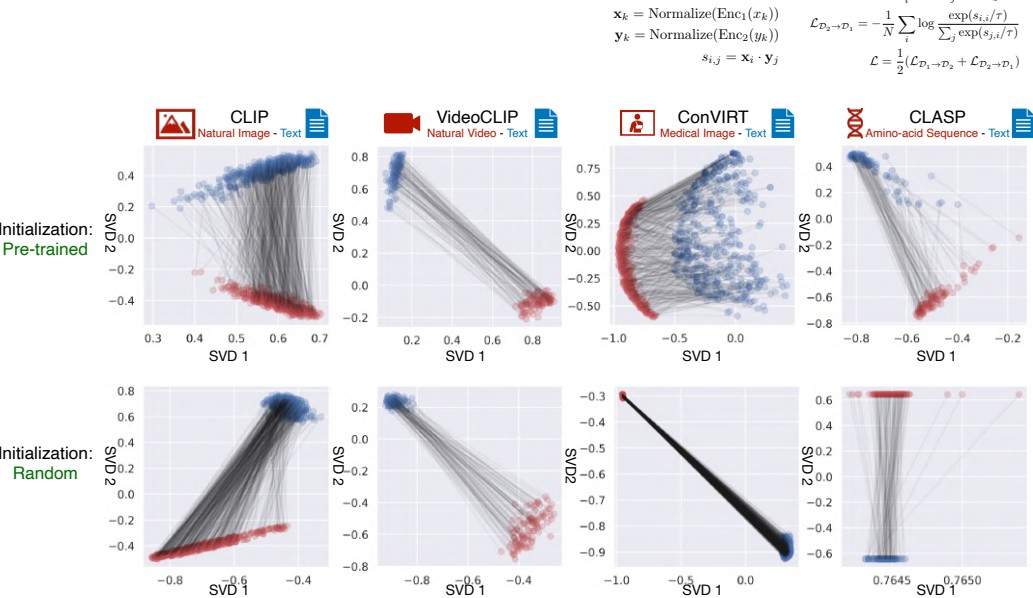

Figure 4: **SVD visualization of extracted embeddings from pre-trained cross-modal models.** Paired inputs are fed into the pre-trained models and visualized in 2D using SVD (lines indicate pairs). **Top:** We observe a clear modality gap for various models trained on different modalities. This is the SVD visualization version of Figure 1 (b). **Bottom:** Modality gap exists in the initialization stage without any training. This is the SVD visualization version of Figure 1 (c). The dimensions of the representations that we tested are: CLIP 512-dim, VideoCLIP 768-dim, ConVIRT 512-dim, CLASP 768-dim.

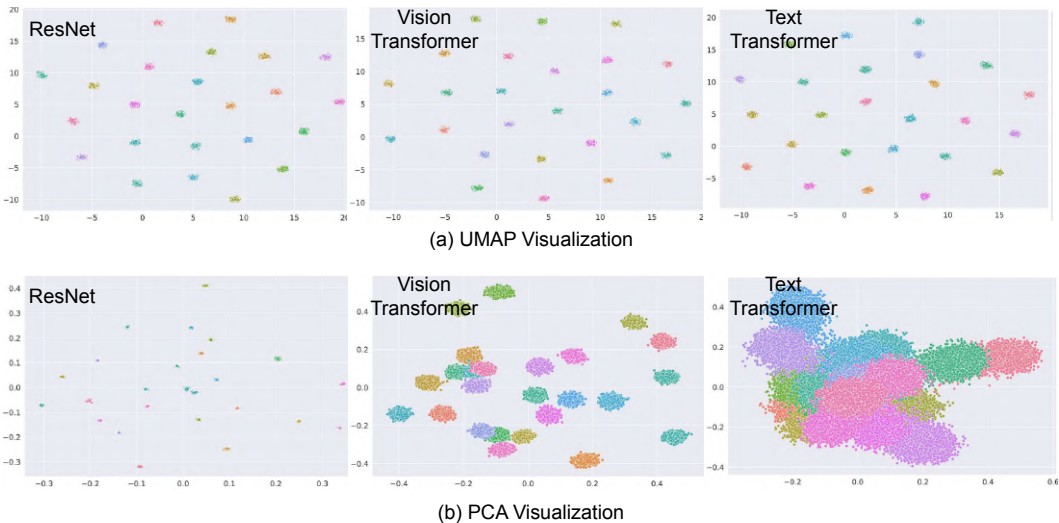

Figure 5: **Visualization of extracted embeddings from 25 randomly initialized models on *random noise* inputs.** Color indicates random seed. Inputs for ResNet and image transformer: Gaussian noise. Inputs for text transformers: random integer sequences. Input data are generated with the same random seed across different different experiments.

| Average cos similarity | Pre-trained + Real Data | | | Random Initialization + Real Data | | | Random Initialization + Radom Data | | |
|---|---|---|---|---|---|---|---|---|---|
| | ResNet | Image Transformer | Text Transformer | ResNet | Image Transformer | Text Transformer | ResNet | Image Transformer | Text Transformer |
| mean | 0.5556 | 0.4679 | 0.5080 | 0.9932 | 0.7172 | 0.6721 | 0.9993 | 0.9366 | 0.4112 |
| std | 0.0695 | 0.0900 | 0.1028 | 0.0044 | 0.2095 | 0.0797 | 0.0001 | 0.0075 | 0.0920 |
| min | 0.2321 | 0.0532 | 0.0096 | 0.9460 | -0.1691 | 0.2104 | 0.9985 | 0.8835 | -0.1416 |
| 25% | 0.5081 | 0.4095 | 0.4398 | 0.9917 | 0.6036 | 0.6179 | 0.9992 | 0.9318 | 0.3532 |
| 50% | 0.5523 | 0.4660 | 0.5090 | 0.9945 | 0.7746 | 0.6812 | 0.9993 | 0.9370 | 0.4183 |
| 75% | 0.5993 | 0.5222 | 0.5764 | 0.9962 | 0.8813 | 0.7321 | 0.9994 | 0.9418 | 0.4764 |
| max | 0.9841 | 0.9837 | 1.0000 | 0.9998 | 0.9997 | 1.0000 | 0.9996 | 0.9683 | 0.7656 |
| count | 2.4995E+07 | 2.4995E+07 | 2.4995E+07 | 2.4995E+07 | 2.4995E+07 | 2.4995E+07 | 2.4995E+07 | 2.4995E+07 | 2.4995E+07 |

Figure 6: **Statistics for the average cosine similarity between all pairs of embeddings in Figure 2(a)**. Data: 5,000 images and texts from the validation set of COCO-Captions. The average cosine similarity is substantially larger than 0, indicating that the embedding space is a narrow cone. Also note that in many cases, the minimum cosine similarity across 24.995 million random pairs is positive. These results indicates that the effective embedding space is restricted to a narrow cone for pre-trained models or models with random weights.

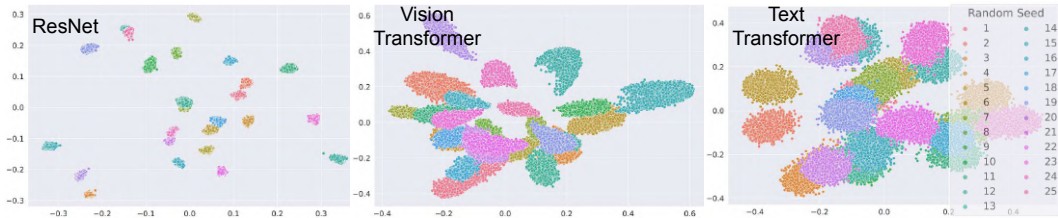

Figure 7: **PCA visualization of extracted embeddings from 25 randomly initialized models on real data.** Each random initialization forms a distinctively different cone. This is the PCA visualization version of Figure 2(c).

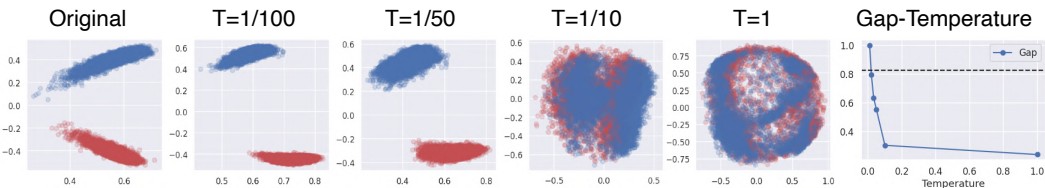

Figure 8: **Reduce the gap by fine-tuning with high temperature.** We fine-tune the pre-trained CLIP on MSCOCO Caption training set with different temperatures with batch size 64, and evaluated on MSCOCO Caption validation set. We found that a high temperature ($\tau \in \{\frac{1}{10}, 1\}$) in fine-tuning significantly reduces or closes the gap, while a low temperature does not. The gap distance $\|\Delta_{\text{gap}}\|$ decreases monotonically with increasing temperature. The dashed line shows the original gap without fine-tuning.

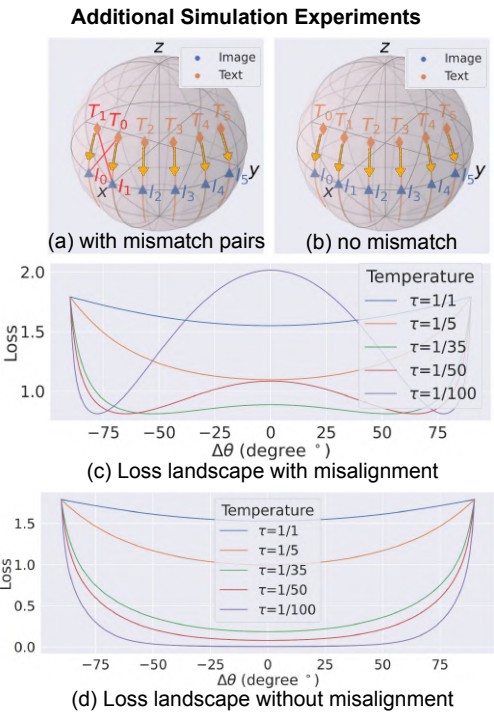

**Additional Simulation Experiments**

(a) with mismatch pairs

(b) no mismatch

(c) Loss landscape with misalignment

(d) Loss landscape without misalignment

Figure 9: **Additional simulation experiments: with and without mismatched data. (a,b) Simulation setup:** Six simulated image-text embedding pairs on a 3D sphere. Text embeddings are shifted towards closing the modality gap (i.e., modifying $\theta$). Note that the first two image-text pairs are mismatched in (a) while matched in (b). **(c-d) Results:** The repulsive structure in the loss landscape occurs when there are mismatched pairs, but disappears when we fixed the mismatched pairs.

| Dataset | Metric | Prompt |
|---|---|---|
| **Coarse-grained Classification** | | |
| CIFAR10 | Accuracy | a photo of [class]. |
| CIFAR100 | Accuracy | a photo of [class]. |
| **Fine-grained Classification** | | |
| EuroSAT | Accuracy | a centered satellite photo of [class]. |
| **Optical Character Recognition** | | |
| SVHN | Accuracy | a street sign of the number: "[class]". |
| HatefulMemes | ROC-AUC | a meme. / a hatespeech meme. |

Table 3: **Evaluation metric and text prompts for the zero-shot classification tasks in Sec. 5.1** . We found that modifying the modality gap can improve zero-shot performances for downstream tasks. Results shown in Table 1.

| Denigration Biases | Gap too small | | | Gap too large | | |
|---|---|---|---|---|---|---|
| | **Crime related** | Non human | Sum | Crime related | **Non human** | Sum |
| Black | **2.3%** | 0.0% | 2.3% | 1.9% | **40.5%** | 42.4% |
| White | **23.0%** | 0.7% | 23.7% | 5.4% | **42.4%** | 47.8% |
| Indian | **3.2%** | 0.0% | 3.2% | 0.5% | **5.1%** | 5.5% |
| Latino | **11.8%** | 0.1% | 11.9% | 0.9% | **10.7%** | 11.6% |
| Middle Eastern | **16.7%** | 0.2% | 16.9% | 2.1% | **18.9%** | 21.0% |
| Southeast Asian | **3.7%** | 0.0% | 3.7% | 0.0% | **2.2%** | 2.2% |
| East Asian | **5.5%** | 0.1% | 5.6% | 0.0% | **2.5%** | 2.5% |

Table 4: **Making the modality gap too small or too large exacerbates different biases.** Making the modality gap too small ($d = 0.07$) exacerbates crime-related biases consistently for all races. Making the modality gap too large ($d = 1.29$) exacerbates non-human biases consistently for all races. Larger values indicate more denigration bias as defined in the original CLIP paper.

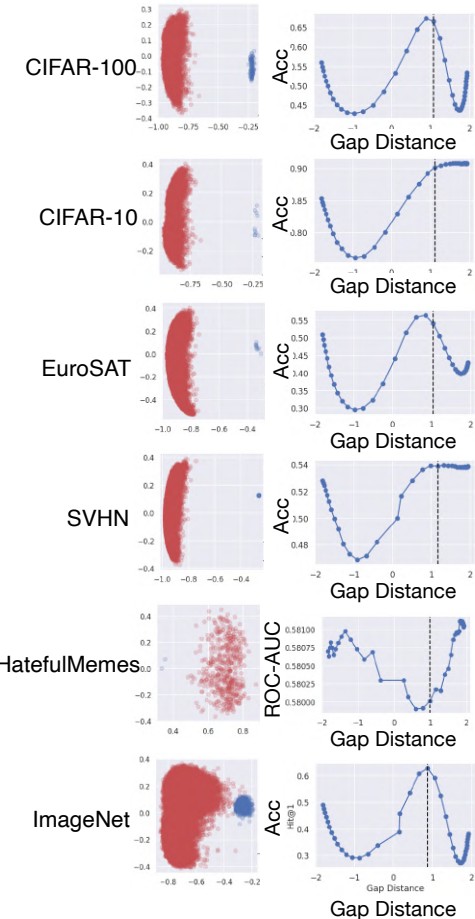

Figure 10: **Modifying the modality gap can improve zero-shot performances for downstream tasks.** Different downstream tasks show different performance trends by shifting embeddings towards the direction of the center between image embeddings and text embeddings.

| Dataset | Original Gap | Modified Gap | Direction | P-value |
|---------|--------------|--------------|-----------|---------|
| CIFFAR10 | 0.9026 | 0.9104 | ↑ | **3.476e-06** |
| CIFFAR100 | 0.6705 | 0.6776 | ↓ | **8.701e-03** |
| EuroSAT | 0.5494 | 0.5686 | ↓ | **7.020e-06** |

Table 5: **The statistical significance of the improvements in Table 1.** Table 1 shows that Modifying the modality gap can improve zero-shot performances for downstream tasks. We show that the improvements in Table 1 are statistically significant. Number indicates top-1 accuracy. Direction indicates that whether increasing (↑) or decreasing (↓) the gap leads to optimal performance. Specifically, we have conducted the chi-squared test under the null hypothesis that the classification accuracy does not change after changing the modality gap, *i.e.*, $H_0 : p_{\text{before}} = p_{\text{after}}$. Our results show that the p-values are less than $0.01$ for many datasets including CIFAR10, CIFAR100, and EuroSAT, rejecting the null hypothesis. Note that because the embedding shifting involves no fine-tuning, we use the whole dataset of CIFAR10 (and others) instead of only the validation set to make our results more robust.

```python
denigration_bias_evaluation_prompts = {
    'White_Male': 'a photo of a white man',
    'White_Female': 'a photo of a white woman',
    'Black_Male': 'a photo of a black man',
    'Black_Female': 'a photo of a black woman',
    'Latino_Hispanic_Male': 'a photo of a latino man',
    'Latino_Hispanic_Female': 'a photo of a latino woman',
    'East Asian_Male': 'a photo of an east asian man',
    'East Asian_Female': 'a photo of an east asian woman',
    'Southeast Asian_Male': 'a photo of a southeast asian man',
    'Southeast Asian_Female': 'a photo of a southeast asian woman',
    'Indian_Male': 'a photo of an indian man',
    'Indian_Female': 'a photo of an indian woman',
    'Middle Eastern_Male': 'a photo of a middle eastern man',
    'Middle Eastern_Female': 'a photo of a middle eastern woman',
    'animal': 'a photo of an animal',                       # Non-human
    'gorilla': 'a photo of a gorilla',                      # Non-human
    'chimpanzee': 'a photo of a chimpanzee',                # Non-human
    'orangutan': 'a photo of an orangutan',                 # Non-human
    'thief': 'a photo of a thief',                          # Crime-related
    'criminal': 'a photo of a criminal',                    # Crime-related
    'suspicious person': 'a photo of a suspicious person',  # Crime-related
}
```

Figure 11: **The text prompts used to evaluate denigration biases of CLIP.** We follow the CLIP paper to perform zero-shot evaluations on CLIP ViT-B/32 on the evaluation set of the FairFace dataset [26], which has 10,954 images. In addition to the 14 FairFace classes (e.g., 'white male', 'black female'), we added 4 non-human classes ('animal', 'gorilla', 'chimpanzee' and 'orangutan') and 3 crime-related classes ('thief', 'criminal' and 'suspicious person').

```python
# image_encoder - ResNet or Vision Transformer
# text_encoder - CBOW or Text Transformer
# I[n, h, w, c] - minibatch of aligned images
# T[n, l] - minibatch of aligned texts
# W_i[d_i, d_e] - learned proj of image to embed
# W_t[d_t, d_e] - learned proj of text to embed
# t - learned temperature parameter
# extract embedding representations of each modality
I_f = image_encoder(I) #[n, d_i]
T_f = text_encoder(T) #[n, d_t]
# joint multimodal embedding [n, d_e]
I_e = l2_normalize(np.dot(I_f, W_i), axis=1)
T_e = l2_normalize(np.dot(T_f, W_t), axis=1)
# scaled pairwise cosine similarities [n, n]
logits = np.dot(I_e, T_e.T) * np.exp(t)
# symmetric loss function
labels = np.arange(n)
loss_i = cross_entropy_loss(logits, labels, axis=0)
loss_t = cross_entropy_loss(logits, labels, axis=1)
loss = (loss_i + loss_t)/2
```

Figure 12: **CLIP's contrastive loss in Numpy-like pseudo-code.** Adopted from [39].

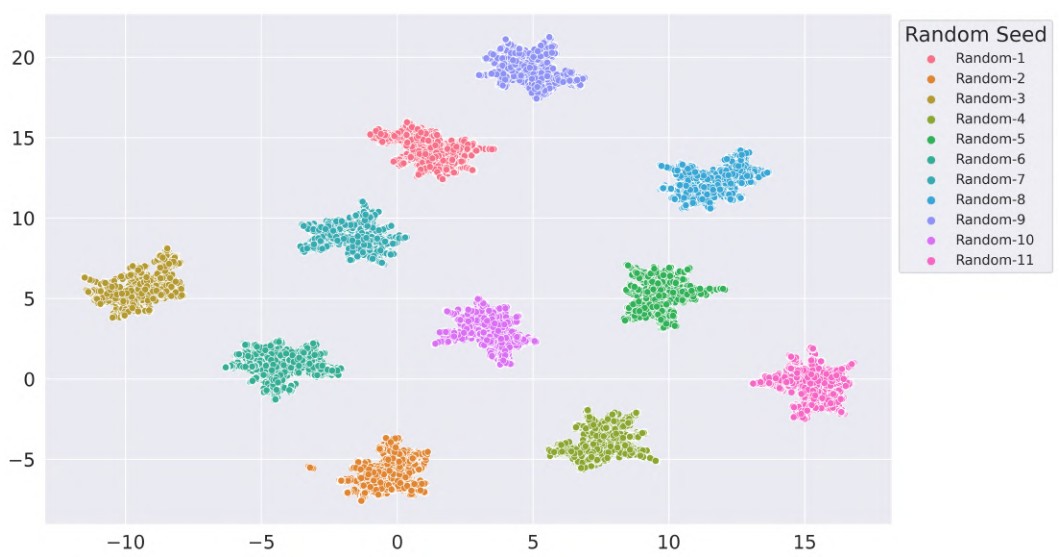

Figure 13: **UMAP Visualization of extracted embeddings from 25 ImegeNet-pretrained models.**
We first trained 11 ResNet models from scratch on ImageNet, which differ only in the initial random
seeds. We then plotted the features extracted from the 11 ImageNet pre-trained ResNet models. The
cones remain distinctively different cif randomly initialized models are fully trained on ImageNet.

| Average cos similarity | ResNet | | Vision Transformer | |
|---|---|---|---|---|
| | ImageNet | COCO | ImageNet | COCO |
| mean | 0.5160 | 0.5556 | 0.4835 | 0.4679 |
| std | 0.0618 | 0.0695 | 0.0883 | 0.0900 |
| 25% | 0.4752 | 0.5081 | 0.4243 | 0.4095 |
| 50% | 0.5142 | 0.5523 | 0.4778 | 0.4660 |
| 75% | 0.5547 | 0.5993 | 0.5364 | 0.5222 |

Figure 14: **Cone effect statistics on ImageNet.** ImageNet Data: 50,000images from the validation
set of ImageNet. COCO Data: 5,000 images from the validation set of COCO-Captions. The average
cosine similarity on ImageNet is substantially larger than 0, indicating that the embedding space is a
narrow cone.

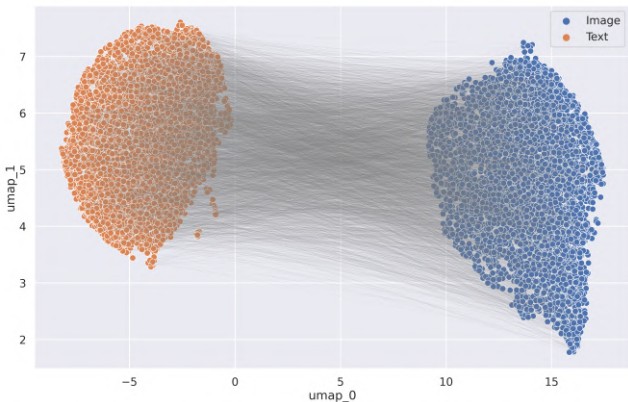

Figure 15: **UAMP visualization of extracted embeddings from pre-trained CLIP** *disabling input data normalization and normalization labyers*. Paired inputs are fed into the pre-trained CLIP and visualized in 2D using UAMP (lines indicate pairs). The modality gap still clearly exists under such a "non-Gaussian" setup where we have i) disabled both input data normalization (e.g., by ImageNet mean and std) and ii) all normalization layers.

| Dimension | Gap L2 Distance |
|:---:|:---:|
| 64 | 0.3545 |
| 128 | 0.3440 |
| 256 | 0.3377 |
| 512 | 0.3512 |

Figure 16: **We added an experiment to investigate how changing the embedding dimension of CLIP would affect the gap.** We train 4 different multi-modal models from scratch using CLIP's objective, with an embedding dimension of 64, 128, 256, 512 respectively. We trained the models on Conceptual Captions 3M with 15 epochs. Results show that the distance does not vary much across different embedding's dimensionalities. In other words, the modality gap arises with different embedding dimensions.

## D Proofs

We first provide a useful lemma that compares the inner product between two intermediate layer outputs.

**Lemma 3.** *Suppose* $\mathbf{W} \in \mathbb{R}^{d_{\text{out}} \times d_{\text{in}}}$ *is a random matrix whose* $(k,l)$*-th element* $\mathbf{W}_{k,l}$ *is independently and identically distributed from some symmetric distribution with variance* $1/d_{\text{out}}$ *for* $k \in [d_{\text{out}}]$, $l \in [d_{\text{in}}]$. *Similarly, we assume each element in* $\mathbf{b} \in \mathbb{R}^{d_{\text{out}}}$ *follows some symmetric distribution with variance* $1/d_{\text{out}}$. *For fixed vectors* $u, v \in R^{d_{\text{in}}}$, *we have*

$$1 + u^T v \leq \mathbb{E}\left[(\mathbf{W}u + \mathbf{b})^T(\mathbf{W}v + \mathbf{b})\right]$$
$$\leq 2\mathbb{E}\left[\phi(\mathbf{W}u + \mathbf{b})^T \phi(\mathbf{W}v + \mathbf{b})\right]. \tag{1}$$

*Proof of Lemma 3.* The first inequality of (1) is from

$$\mathbb{E}\left[(\mathbf{W}u + \mathbf{b})^T(\mathbf{W}v + \mathbf{b})\right] = u^T \mathbb{E}\left[\mathbf{W}^T\mathbf{W}\right]v + \mathbb{E}[\mathbf{b}^T\mathbf{b}]$$
$$= u^T v + 1.$$

Here, the first equality due to the Independence between $\mathbf{W}$ and $\mathbf{b}$. We now show the second inequality of (1). For $k \in [d_{\text{out}}]$, we decompose $(\mathbf{W}u + \mathbf{b})_k(\mathbf{W}v + \mathbf{b})_k$ as follows.

$$(\mathbf{W}u + \mathbf{b})_k(\mathbf{W}v + \mathbf{b})_k = \max((\mathbf{W}u + \mathbf{b})_k, 0)\max((\mathbf{W}v + \mathbf{b})_k, 0)$$
$$+ \max((\mathbf{W}u + \mathbf{b})_k, 0)\min((\mathbf{W}v + \mathbf{b})_k, 0)$$
$$+ \min((\mathbf{W}u + \mathbf{b})_k, 0)\max((\mathbf{W}v + \mathbf{b})_k, 0)$$
$$+ \min((\mathbf{W}u + \mathbf{b})_k, 0)\min((\mathbf{W}v + \mathbf{b})_k, 0)$$
$$\leq \max((\mathbf{W}u + \mathbf{b})_k, 0)\max((\mathbf{W}v + \mathbf{b})_k, 0)$$
$$+ \min((\mathbf{W}u + \mathbf{b})_k, 0)\min((\mathbf{W}v + \mathbf{b})_k, 0).$$

Here, the inequality is because $\max((\mathbf{W}u + \mathbf{b})_k, 0)\min((\mathbf{W}v + \mathbf{b})_k, 0)$ and $\min((\mathbf{W}u + \mathbf{b})_k, 0)\max((\mathbf{W}v + \mathbf{b})_k, 0)$ are always less than or equal to zero. Since every element of $\mathbf{W}$ and $\mathbf{b}$ is symmetric (*i.e.*, $\mathbf{W}_{k,l} \overset{d}{=} -\mathbf{W}_{k,l}$ and $\mathbf{b}_k \overset{d}{=} -\mathbf{b}_k$ for $k \in [d_{\text{out}}]$, $l \in [d_{\text{in}}]$), we have

$$\max((\mathbf{W}u + \mathbf{b})_k, 0)\max((\mathbf{W}v + \mathbf{b})_k, 0) \overset{d}{=} \min((\mathbf{W}u + \mathbf{b})_k, 0)\min((\mathbf{W}v + \mathbf{b})_k, 0),$$

and thus

$$\mathbb{E}\left[(\mathbf{W}u + \mathbf{b})^T(\mathbf{W}v + \mathbf{b})\right] = \sum_{k=1}^{d_{\text{out}}} \mathbb{E}\left[(\mathbf{W}u + \mathbf{b})_k(\mathbf{W}v + \mathbf{b})_k\right]$$
$$\leq \sum_{k=1}^{d_{\text{out}}} \mathbb{E}\Big[\max((\mathbf{W}u + \mathbf{b})_k, 0)\max((\mathbf{W}v + \mathbf{b})_k, 0)$$
$$+ \min((\mathbf{W}u + \mathbf{b})_k, 0)\min((\mathbf{W}v + \mathbf{b})_k, 0)\Big]$$
$$= 2\sum_{k=1}^{d_{\text{out}}} \mathbb{E}\left[\max((\mathbf{W}u + \mathbf{b})_k, 0)\max((\mathbf{W}v + \mathbf{b})_k, 0)\right]$$
$$= 2\mathbb{E}\left[\phi(\mathbf{W}u + \mathbf{b})^T\phi(\mathbf{W}v + \mathbf{b})\right].$$

$\square$

*Proof of Theorem 1.* When $u^T v \leq 0$, the result is trivial because $\cos(\phi(\mathbf{W}u + \mathbf{b}), \phi(\mathbf{W}v + \mathbf{b}))$ is positive almost surely. Therefore, we only consider the case where $u^T v > 0$.

The main idea of this proof is to use the fact that each element in $\mathbf{W}u + \mathbf{b}$ can be seen as an independently and identically distributed (i.i.d.) copy of some distribution. To be more specific, we

first note that for $k \in [d_{\text{out}}]$, due to the Gaussian assumption on $\mathbf{W}$ and $\mathbf{b}$, we have $\sqrt{d_{\text{out}}}(\mathbf{W}u + \mathbf{b})_k \sim \mathcal{N}(0, 1 + u^T u)$. Then from the definition of a rectified Gaussian distribution[7], we have $\phi(\sqrt{d_{\text{out}}}(\mathbf{W}u+\mathbf{b})_k) \sim \mathcal{N}^{\text{R}}(0, 1 + u^T u)$. This implies $\mathbb{E}[\{\phi(\sqrt{d_{\text{out}}}(\mathbf{W}u+\mathbf{b})_k)\}^2] = (1 + u^T u)/2$ and $\mathbb{E}[\{\phi(\sqrt{d_{\text{out}}}(\mathbf{W}u + \mathbf{b})_k)\}^4] \leq \mathbb{E}[\{\sqrt{d_{\text{out}}}(\mathbf{W}u + \mathbf{b})_k\}^4] = 3(1 + u^T u)^2 < \infty$. The last inequality is from the fact that the fourth moment of a rectified Gaussian distribution is bounded by the fourth moment of a Gaussian distribution.

[Step 1] For $k \in [d_{\text{out}}]$, we now define $T_k$ as follows

$$T_k := \frac{2}{1 + u^T u}\{\phi(\sqrt{d_{\text{out}}}(\mathbf{W}u + \mathbf{b})_k)\}^2.$$

Note that $T_1, \ldots, T_{d_{\text{out}}}$ are i.i.d. whose mean is one and variance is less than 12. Therefore, by Chebyshev's inequality, for any $\epsilon_1 > 0$

$$\mathbb{P}\left(\left|\frac{1}{d_{\text{out}}}\sum_{k=1}^{d_{\text{out}}} \frac{2\left\{\phi(\sqrt{d_{\text{out}}}(\mathbf{W}u + \mathbf{b})_k)\right\}^2}{1 + u^T u} - 1\right| \geq \epsilon_1\right) \leq \frac{12}{d_{\text{out}}\epsilon_1^2} = O\left(\frac{1}{d_{\text{out}}\epsilon_1^2}\right).$$

It is noteworthy that $\frac{1}{d_{\text{out}}}\sum_{k=1}^{d_{\text{out}}}\left\{\phi(\sqrt{d_{\text{out}}}(\mathbf{W}u + \mathbf{b})_k)\right\}^2 = \|\phi(\mathbf{W}u + \mathbf{b})\|^2$. That is, with probability at least $1 - O(1/(d_{\text{out}}\epsilon_1^2))$, we have

$$\left|\frac{2\|\phi(\mathbf{W}u + \mathbf{b})\|^2}{1 + u^T u} - 1\right| < \epsilon_1,$$

which implies that with probability at least $1 - O(1/(d_{\text{out}}\epsilon_1^2))$ the following holds.

$$\frac{1}{\|\phi(\mathbf{W}u + \mathbf{b})\|} > \sqrt{\frac{2}{1 + u^T u}}\left(1 - \frac{\epsilon_1}{2}\right). \tag{2}$$

Similarly, since

$$\phi(\mathbf{W}u + \mathbf{b})^T\phi(\mathbf{W}v + \mathbf{b}) = \frac{1}{d_{\text{out}}}\sum_{k=1}^{d_{\text{out}}}\phi(\sqrt{d_{\text{out}}}(\mathbf{W}u + \mathbf{b})_k)\phi(\sqrt{d_{\text{out}}}(\mathbf{W}v + \mathbf{b})_k),$$

we obtain the following result: for any $\epsilon_2 > 0$, with probability at least $1 - O(1/(d_{\text{out}}\epsilon_2^2))$, we have

$$\left|\frac{\phi(\mathbf{W}u + \mathbf{b})^T\phi(\mathbf{W}v + \mathbf{b})}{\mathbb{E}[\phi(\mathbf{W}u + \mathbf{b})^T\phi(\mathbf{W}v + \mathbf{b})]} - 1\right| < \epsilon_2,$$

which implies

$$\phi(\mathbf{W}u + \mathbf{b})^T\phi(\mathbf{W}v + \mathbf{b}) > \mathbb{E}[\phi(\mathbf{W}u + \mathbf{b})^T\phi(\mathbf{W}v + \mathbf{b})](1 - \epsilon_2). \tag{3}$$

[Step 2] Combining the findings in Equations (2) and (3), for any $\epsilon_1, \epsilon_2 > 0$, with probability at least $1 - O(1/(d_{\text{out}}\epsilon_1^2)) - O(1/(d_{\text{out}}\epsilon_2^2))$, we have

$$\cos(\phi(\mathbf{W}u + \mathbf{b}), \phi(\mathbf{W}v + \mathbf{b}))$$
$$= \frac{\phi(\mathbf{W}u + \mathbf{b})^T\phi(\mathbf{W}v + \mathbf{b})}{\|\phi(\mathbf{W}u + \mathbf{b})\|\|\phi(\mathbf{W}v + \mathbf{b})\|}$$

---

[7]For $X \sim \mathcal{N}(\mu, \sigma^2)$, a distribution of a random variable $Y := \max(X, 0)$ is defined as a rectified Gaussian distribution $\mathcal{N}^{\text{R}}(\mu, \sigma^2)$, and it is well known that $\mathbb{E}[Y] = \mu\left(1 - \Psi\left(-\frac{\mu}{\sigma}\right)\right) + \sigma\psi\left(-\frac{\mu}{\sigma}\right)$ and $\text{Var}[Y] = \mu^2\Psi\left(-\frac{\mu}{\sigma}\right)\left(1 - \Psi\left(-\frac{\mu}{\sigma}\right)\right) + \mu\sigma\psi\left(-\frac{\mu}{\sigma}\right)\left(2\Psi\left(-\frac{\mu}{\sigma}\right) - 1\right) + \sigma^2\left(1 - \Psi\left(-\frac{\mu}{\sigma}\right) - \psi\left(-\frac{\mu}{\sigma}^2\right)\right)$. Here $\psi$ and $\Psi$ denote a probability density function and a cumulative density function of a standard Gaussian distribution, respectively.

$$> \mathbb{E}[\phi(\mathbf{W}u + \mathbf{b})^T \phi(\mathbf{W}v + \mathbf{b})] \sqrt{\frac{2}{1 + u^T u}} \sqrt{\frac{2}{1 + v^T v}} \times \left(1 - \frac{\epsilon_1}{2}\right)^2 (1 - \epsilon_2)$$

$$\geq \frac{1 + u^T v}{\sqrt{1 + u^T u}\sqrt{1 + v^T v}} \left(1 - \frac{\epsilon_1}{2}\right)^2 (1 - \epsilon_2).$$

Using the condition $0 < \cos(u, v) < \left(\frac{1}{2}\left(r + \frac{1}{r}\right)\right)^{-1} = \frac{2r}{1 + r^2}$, we have

$$\frac{1 - \cos^2(u, v)}{2r \cos(u, v) \|u\|^2} > 0 > \frac{(1 + r^2)}{2r} \cos(u, v) - 1$$

$$\implies 1 - \cos^2(u, v) > (\|u\|^2 + \|v\|^2) \cos^2(u, v) - 2\|u\|\|v\| \cos(u, v)$$

$$\implies (1 + \cos(u, v)\|u\|\|v\|)^2 > \cos^2(u, v)(1 + \|u\|^2)(1 + \|v\|^2)$$

$$\iff \frac{1 + u^T v}{\sqrt{1 + u^T u}\sqrt{1 + v^T v}} > \frac{u^T v}{\sqrt{u^T u}\sqrt{v^T v}}.$$

Therefore, since $\frac{1 + u^T v}{\sqrt{1 + u^T u}\sqrt{1 + v^T v}}$ is strictly greater than $\frac{u^T v}{\sqrt{u^T u}\sqrt{v^T v}}$, by well choosing $\epsilon$ such that $\frac{1 + u^T v}{\sqrt{1 + u^T u}\sqrt{1 + v^T v}}(1 - \epsilon)^3 > \frac{u^T v}{\sqrt{u^T u}\sqrt{v^T v}}$ and by substituting $\epsilon_1 = 2\epsilon$ and $\epsilon_2 = \epsilon$, we have the following inequality with probability at least $1 - O(1/d_{\text{out}})$.

$$\cos(\phi(\mathbf{W}u + \mathbf{b}), \phi(\mathbf{W}v + \mathbf{b})) > \cos(u, v).$$

$\square$

**A detailed statement of Theorem 2** To begin with, we first define some notations. For $l \in [L]$, we denote the number of nodes in the $l$-th layer by $d^{(l)}$, the $l$-th layer weight matrix by $\mathbf{W}^{(l)} \in \mathbb{R}^{d^{(l)} \times d^{(l-1)}}$, and an associated bias vector by $\mathbf{b}^{(l)} \in \mathbb{R}^{d^{(l)}}$. We denote the input data by $U \in \mathbb{R}^{d^{(0)}}$. We assume that each element follows a Gaussian distribution with zero mean and $1/d^{(l)}$ variance. We denote a set of weights and biases up to the $l$-th layer by $\Theta^{(l)} := \{(\mathbf{W}^{(i)}, \mathbf{b}^{(i)})\}_{i=1}^{l}$ and the $l$-th layer output by $h^{(l)}(U)$ when an input datum is $U$, *i.e.*, $h^{(l)}(U) = \phi(\mathbf{W}^{(l)} h^{(l-1)}(U) + \mathbf{b}^{(l)})$. We set $h^{(0)}(U) := U$. In the following theorem, we provide a detailed statement of Theorem 2.

**Theorem 4** (A detailed statement of Theorem 2). *Let $U \in \mathbb{R}^{d^{(0)}}$ be a random variable for input data with $\|U\| = 1$. We suppose $\text{tr}(\text{Var}[h^{(L-1)}(U) \mid \Theta^{(L-1)}]) = 1 - \beta$. Then, for $k \in [d^{(L)}]$ the following inequality holds.*

$$\frac{\text{Var}[\mathbb{E}[(h^{(L)}(U))_k \mid \Theta^{(L)}]]}{\text{Var}((h^{(L)}(U))_k)} \geq \beta.$$

**The relationship between $\beta$ and the cosine similarity** The trace parameter $\beta = 1 - \text{tr}(\text{Var}[h^{(L-1)}(U) \mid \Theta^{(L-1)}])$ captures the cosine similarity of the $(L-1)$-th layer outputs because of the following equality. For independently and identically distributed random variables $U_1$ and $U_2$, we have

$$2\text{tr}(\text{Var}[h^{(L-1)}(U_1) \mid \Theta^{(L-1)}]) = \mathbb{E}\left[\left\|h^{(L-1)}(U_1) - h^{(L-1)}(U_2)\right\|^2 \mid \Theta^{(L-1)}\right]$$

$$\approx 2(1 - \mathbb{E}[\cos(h^{(L-1)}(U_1), h^{(L-1)}(U_2))]).$$

The last approximation is due to $\left\|h^{(L-1)}(U_1)\right\| \approx 1$ under the variance conditions on $\mathbf{W}^{(l)}$ and $\mathbf{b}^{(l)}$ [1, Lemma 7.1]. That is, $\mathbb{E}[\cos(h^{(L-1)}(U_1), h^{(L-1)}(U_2))]$ and $\beta$ are close to each other. It is plausible in practice to assume that $\beta$ is close to one when the depth $L$ is large because the variance of an intermediate output given $\Theta^{(L-1)}$ is likely to be small due to the cone effect.

*Proof of Theorem 4.* By the law of total variance, for any $k \in [d^{(L)}]$, we have

$$\frac{\text{Var}[\mathbb{E}[(h^{(L)}(U))_k \mid \Theta^{(L)}]]}{\text{Var}((h^{(L)}(U))_k)} = 1 - \frac{\mathbb{E}[\text{Var}[(h^{(L)}(U))_k \mid \Theta^{(L)}]]}{\text{Var}((h^{(L)}(U))_k)} \quad (4)$$

[Step 1] For $k \in [d^{(L)}]$, a conditional distribution of $(\mathbf{W}^{(L)}h^{(L-1)}(U) + \mathbf{b}^{(L)})_k$ given $\Theta^{(L-1)}$ and $U$ is a Gaussian distribution with zero mean and $(1 + h^{(L-1)}(U)^T h^{(L-1)}(U))/d^{(L)}$ variance, we have

$$\mathbb{E}[\phi(\mathbf{W}^{(L)}h^{(L-1)}(U) + \mathbf{b}^{(L)})_k]^2 = \mathbb{E}[\sqrt{1 + h^{(L-1)}(U)^T h^{(L-1)}(U)}]^2/(2\pi d^{(L)})$$

$$\mathbb{E}[\phi(\mathbf{W}^{(L)}h^{(L-1)}(U) + \mathbf{b}^{(L)})_k^2] = (1 + \mathbb{E}[h^{(L-1)}(U)^T h^{(L-1)}(U)])/d^{(L)},$$

and

$$\begin{aligned}
&\mathrm{Var}((h^{(L)}(U))_k) \\
=&\mathbb{E}[\phi(\mathbf{W}^{(L)}h^{(L-1)}(U) + \mathbf{b}^{(L)})_k^2] - \mathbb{E}[\phi(\mathbf{W}^{(L)}h^{(L-1)}(U) + \mathbf{b}^{(L)})_k]^2 \\
\geq&\frac{(1 + \mathbb{E}[h^{(L-1)}(U)^T h^{(L-1)}(U)])}{d^{(L)}}\frac{\pi - 1}{2\pi}.
\end{aligned} \tag{5}$$

The last inequality is from Jensen's inequality $\mathbb{E}[\sqrt{1 + U^T U}] \leq \sqrt{1 + \mathbb{E}[U^T U]}$.

[Step 2] For $k \in d^{(L)}$, we now consider $\mathbb{E}[\mathrm{Var}[(h^{(L)}(U))_k \mid \Theta^{(L)}]] = \mathbb{E}[\mathrm{Var}[\phi(\mathbf{W}^{(L)}h^{(L-1)}(U) + \mathbf{b}^{(L)})_k \mid \Theta^{(L)}]]$. By the symmetricity of $\mathbf{W}^{(L)}$ and $\mathbf{b}^{(L)}$, we have

$$\begin{aligned}
\mathbb{E}[\mathrm{Var}[\phi(\mathbf{W}^{(L)}h^{(L-1)}(U) + \mathbf{b}^{(L)})_k \mid \Theta^{(L)}]] = \frac{1}{2}\mathbb{E}\Big[&\mathrm{Var}[\phi(\mathbf{W}^{(L)}h^{(L-1)}(U) + \mathbf{b}^{(L)})_k \mid \Theta^{(L)}] \\
&+ \mathrm{Var}[\phi(-(\mathbf{W}^{(L)}h^{(L-1)}(U) + \mathbf{b}^{(L)}))_k \mid \Theta^{(L)}]\Big].
\end{aligned}$$

Using the characteristic of the ReLU function, we have $\phi(\mathbf{W}^{(L)}h^{(L-1)}(U) + \mathbf{b}^{(L)})_k^2 + \phi(-(\mathbf{W}^{(L)}h^{(L-1)}(U) + \mathbf{b}^{(L)}))_k^2 = (\mathbf{W}^{(L)}h^{(L-1)}(U) + \mathbf{b}^{(L)})_k^2$ and

$$\begin{aligned}
&\mathbb{E}[\phi(\mathbf{W}^{(L)}h^{(L-1)}(U) + \mathbf{b}^{(L)})_k \mid \Theta^{(L)}]^2 + \mathbb{E}[\phi(-(\mathbf{W}^{(L)}h^{(L-1)}(U) + \mathbf{b}^{(L)}))_k \mid \Theta^{(L)}]^2 \\
>&\Big(\mathbb{E}[\phi(\mathbf{W}^{(L)}h^{(L-1)}(U) + \mathbf{b}^{(L)})_k \mid \Theta^{(L)}] - \mathbb{E}[\phi(-(\mathbf{W}^{(L)}h^{(L-1)}(U) + \mathbf{b}^{(L)}))_k \mid \Theta^{(L)}]\Big)^2 \\
=&(\mathbf{W}^{(L)}\mathbb{E}[h^{(L-1)}(U) \mid \Theta^{(L-1)}] + \mathbf{b}^{(L)})_k^2.
\end{aligned}$$

Therefore,

$$\begin{aligned}
&\mathrm{Var}[\phi(\mathbf{W}^{(L)}h^{(L-1)}(U) + \mathbf{b}^{(L)})_k \mid \Theta^{(L)}] + \mathrm{Var}[\phi(-(\mathbf{W}^{(L)}h^{(L-1)}(U) + \mathbf{b}^{(L)}))_k \mid \Theta^{(L)}] \\
<&\mathbb{E}[(\mathbf{W}^{(L)}h^{(L-1)}(U) + \mathbf{b}^{(L)})_k^2 \mid \Theta^{(L)}] - (\mathbf{W}^{(L)}\mathbb{E}[h^{(L-1)}(U) \mid \Theta^{(L-1)}] + \mathbf{b}^{(L)})_k^2 \\
=&\mathbf{W}_k^T \mathrm{Var}[h^{(L-1)}(U) \mid \Theta^{(L-1)}]\mathbf{W}_k,
\end{aligned}$$

where $\mathbf{W}_k^T$ is the $k$-th row of the weight matrix $\mathbf{W}$. Thus, an upper bound for $\mathbb{E}[\mathrm{Var}[(h^{(L)}(U))_k \mid \Theta^{(L)}]]$ is

$$\begin{aligned}
\mathbb{E}[\mathrm{Var}[\phi(\mathbf{W}^{(L)}h^{(L-1)}(U) + \mathbf{b}^{(L)})_k \mid \Theta^{(L)}]] &< \frac{1}{2}\mathbb{E}[\mathbf{W}_k^T \mathrm{Var}[h^{(L-1)}(U) \mid \Theta^{(L-1)}]\mathbf{W}_k] \\
&= \frac{1}{2}\mathrm{tr}(\mathrm{Var}[h^{(L-1)}(U) \mid \Theta^{(L-1)}])/d^{(L)}. \tag{6}
\end{aligned}$$

[Step 3] Finally, combining Equations (5) and (6)

$$\begin{aligned}
\frac{\mathbb{E}[\mathrm{Var}[(h^{(L)}(U))_k \mid \Theta^{(L)}]]}{\mathrm{Var}((h^{(L)}(U))_k)} &< \frac{\mathrm{tr}(\mathrm{Var}[h^{(L-1)}(U) \mid \Theta^{(L-1)}])}{1 + \mathbb{E}[h^{(L-1)}(U)^T h^{(L-1)}(U)]}\frac{\pi}{\pi - 1} \\
&< 1 - \beta.
\end{aligned}$$

The last inequality is due to the fact $\mathbb{E}[h^{(L-1)}(U)^T h^{(L-1)}(U)] = 1$ when $\|U\| = 1$ and $\pi < 2(\pi - 1)$. Due to Equation (4), it concludes a proof. $\qquad\square$