# OpenReview forum: "Mind the Gap: Understanding the Modality Gap in Multi-modal Contrastive Representation Learning"
_NeurIPS.cc/2022/Conference — NeurIPS 2022 Accept_

### Official Review · Reviewer_QxpT · 2022-07-09

**Rating:** 7
**Confidence:** 5
**Soundness:** 3 good
**Presentation:** 4 excellent
**Contribution:** 3 good

**Summary:**

This paper investigates the modality gap issue in existing multi-modal models. Specifically, this work empirically shows the gap phenomenon across different neural networks and different modalities, and makes a conclusion that the gap issue is a combination of model initialization and contrastive learning optimization. This paper further theoretically proves how inductive bias in deep neural networks creates narrow representation cones in the embedding space. Another contribution of this work is to empirically investigate the impact of the modality gap distance in downstream tasks.


**Questions:**

1. What’s the zero-shot performance on ImageNet?
2. If the CLIP model is pre-trained on large-scale datasets, does the modality gap still have an impact on the downstream tasks?


**Strengths And Weaknesses:**

---

Originality: to the best of my knowledge, this is the first work that investigates the modality gap of multi-modal models. It is interesting to see that neural networks with random initialization create narrow cones in the embedding space. This paper also theoretically proves how it happens.

---

Quality:

Strengths: the empirical study and theoretical proof of the cone effect in deep neural networks are solid and well supported. Specifically, this work first empirically demonstrates that the cone effect widely exists in deep neural networks, then theoretically prove that each network layer narrows the representation cone.

Weakness: (1) as shown in Table 1, the impact of modality gap on downstream tasks is marginal. (2) All experiments are performed on small-scale datasets. For example, the pretraining is performed on MSCOCO, and downstream tasks are performed on CIFAR10, CIFAR100, and SVHN. What’s the zero-shot performance on ImageNet? If the CLIP model pretrained on large-scale datasets, does the modality gap still have an impact on the downstream tasks?

---

Clarity: this paper is well written and well organized. It is easy to follow. Since this is an empirical work and built upon existing models, there is no reproduction issue.

---

Significance: the investigation of the modality gap in multi-modal models is interesting and would benefit the community.

---

---

> ### Author Response · Authors · 2022-08-02
> **Response to Reviewer QxpT**
>
> We thank Reviewer QxpT for their positive comments and for providing thoughtful feedback on our work. We address many of Reviewer QxpT’s comments in our general response above, and we provide additional details on specific comments below.
>
> **The pre-training data of CLIP**
> > For example, the pretraining is performed on MSCOCO
> > If the CLIP model is pre-trained on large-scale datasets, does the modality gap still have an impact on the downstream tasks?
>
> Thanks for the question. We clarified that the CLIP model we study is pre-trained on the original OpenAI image-caption dataset, not the MSCOCO dataset. The OpenAI image-caption dataset contains 400 million image-caption pairs, which is a very large-scale dataset.
>
>
> **Improvements shown in Table 1**
> > as shown in Table 1, the impact of modality gap on downstream tasks is marginal.
>
> Thank you for the comment. We have shown that the improvements are statistically significant with p-value<0.01, and we refer the reviewer to the general response above. We have added the statistical testing to page 19 of the revised paper (See Supplementary Material). We also reiterate that the main objective of our paper is to understand the modality gap phenomenon, a general inductive bias that holds across various data modalities and NN architectures. The goal of our paper is not to propose a method to close the gap or to improve downstream performance, which is an important direction of follow-up work.
>
>
>
> **Zero-shot classification experiment on ImageNet**
> > What’s the zero-shot performance on ImageNet?
>
> Thank you for your comment. We have added new zero-shot experiments on ImageNet. New results are shown on page 19 of the revised paper (See Supplementary Material).
>
> We found that making the gap smaller or larger by feature shifting decreases the model performance on ImageNet. Although changing the gap does not improve the model performance, and it is not clear why the model performs best with the default gap, we can still clearly see the effect of the modality gap on zero-shot classification.
>
> Moreover, we reiterate that the main objective of our paper is to i) empirically demonstrate the modality gap phenomenon across different data modalities and NN architectures; ii) explain how the gap arises, and iii) show that the size of the gap can affect downstream applications. It is not our goal to propose a method to close the gap, since it’s not clear that it’s desirable to have no modality gap.

---

> ### Author Response · Authors · 2022-08-09
> **We would like to hear back from Reviewer QxpT**
>
> Dear reviewer QxpT,
>
> We would like to follow up to see if our response addresses your concerns or if you have any further questions. We would really appreciate the opportunity to discuss this further if our response has not already addressed your concerns. Thank you very much!

---

### Official Review · Reviewer_nReN · 2022-07-10

**Rating:** 6
**Confidence:** 3
**Soundness:** 3 good
**Presentation:** 3 good
**Contribution:** 3 good

**Summary:**

The paper proposes ‘modality gap’ in multimodal representation learning. The authors observe that embeddings of different modalities under a CLIP-style contrastive training objective will fall in different regions of the embedding space. They further provide explanations for the modality gap and a theoretical analysis.


**Questions:**

+ Figure 3, the objective of contrastive learning is cosine similarity while Figure 3 plots Euclidean distance. Could the authors provide plots for the cosine similarity or comment on reasons for using Euclidean distance? Is is possible that the embedding shift is related to difference between cosine distance and Euclidean distance?

+ Is the proposed modality gap specific to multi-modal learning? Does the gap exist for unimodal contrastive learning?

+ Section 4.3 (simulating mismatched data) seems an interesting investigation. What’s the multi-modal data being used here? Also, I wonder if the authors have some insight on how the modality gap correlates with multimodal data misalignment (e.g., an image is paired with a wrong caption due to data collection error).

+ The improvement in Table 1 seems quite small. Is the reported number given by exhaustively iterating over all possible shifting? Are the results averaged over multiple runs / random seeds?

+ Line309: shwon -> shown

**Limitations:**


As the authors mention in the paper, an important future direction is to investigate how the modality gap influence downstream task performance.

**Strengths And Weaknesses:**

\+ The paper presents an interesting phenomenon and unique insight in multimodal representation learning

\+ The conclusion is evaluated with many experiments and backed with theoretical analysis.

\+ The paper is well written and clearly organized.

&nbsp;


\- What’s unclear to me is that: modality gap seems to stem from the inductive bias of deep neural networks. The key is that using two encoders (with different initialization) to process input data will produce a gap in the representation space, and a contrastive learning objective preserves the gap. I don’t quite see the influence of data modality in the problem statement. As stated in L108-110, the gap exists even if the two encoders operate on the exact same data (i.e., no multi modality). If so, why is the gap termed as ‘modality gap’? What differentiates the gap in a multi-modal learning setting from a uni-modal setting?

If we consider a uni-modal setting, take Mean Teacher [1] in semi-supervised learning as an example: given an image, the consistency loss aims to minimize the predictions of a teacher and student model. The teacher model is the exponential moving average of the student model weights (i.e., the two models have different weights and operate on same data modality). Does the gap still exist here if we use a contrastive learning objective?

Conversely, if we use a single model for multiple modalities (e.g., Omnivore [2]), where different modalities are tokenized and fed into the same transformer. Will the modality gap still exist here?

[1] Mean teachers are better role models: Weight-averaged consistency targets improve semi-supervised deep learning results.
[2] Omnivore: A Single Model for Many Visual Modalities

---

> ### Author Response · Authors · 2022-08-02
> **Response to Reviewer nReN**
>
> We thank Reviewer nReN for their positive comments and for providing thoughtful feedback on our work. We address many of Reviewer nReN’s comments in our general response above, and we provide additional details on specific comments below.
>
> **Why Euclidean distance to quantify the gap**
> > Figure 3, the objective of contrastive learning is cosine similarity while Figure 3 plots Euclidean distance. Could the authors provide … comment on reasons for using Euclidean distance?
> > Is it possible that the embedding shift is related to difference between cosine distance and Euclidean distance?
>
> Thank you for the question. In CLIP, the image embeddings and text embeddings are L2-normalized (See Supplementary Figure 12: CLIP’s contrastive loss in Numpy-like pseudo-code). In other words, the image and text embeddings of CLIP are always on the $n$-dimensional unit sphere (n=512). Specifically, for any $n$-dimensional vectors $x$ and $y$, the cosine similarity is given as $\cos(x,y)=x^T y$, and the Euclidean distance is given as $(x-y)^T (x-y) = 2(1-x^T y)$. Therefore, they have a functional relationship as $\mathrm{Euclidean Distance}(x,y)=2(1-\cos(x,y))$. When the angle between $x$ and $y$ is less than $\pi/2$, which is the case as embeddings are in a narrow cone, the small Euclidean distance directly means a high cosine similarity.
>
>
> We have added these clarifications in the caption of Figure 3 and Section 4.2.
>
>
> **Why is the gap termed as ‘modality gap’**
> > The gap exists even if the two encoders operate on the exact same data (i.e., no multi modality). If so, why is the gap termed as ‘modality gap’? What differentiates the gap in a multi-modal learning setting from a uni-modal setting?
>
> Thank you for the comment. We term the gap as “modality gap” because the main focus of our paper is in multi-model contrastive representation learning, which is an important research area that has garnered tremendous interest and excitement. We observe two modality clusters in various multi-modality learning settings in Figure 1 that seemingly capture differences in modalities. That is the main reason why we call this gap the modality gap, but our in-depth analyses show that there are many factors leading to such gaps. This counterintuitive conclusion is exactly why we believe our finding is surprising and of great interest to the multimodal learning community.
>
>
>
> **Modality gap with only one shared encoder**
> > Conversely, if we use a single model for multiple modalities (e.g., Omnivore [2]), where different modalities are tokenized and fed into the same transformer. Will the modality gap still exist here?
>
> Thank you for the question. Multimodal contrastive representation learning involves two separate encoders, which create two different cones in the representation space. In contrast, there is only one encoder in Omnivore [5]. Our analyses assumed two separate encoders, and thus do not directly generalize to the one encoder case.
>
> Because Omnivore [5] has only one shared encoder, different modalities would be mapped into the same cone of the shared encoder. Exploring the geometry within that cone, especially whether different modalities are located in completely separate regions within that cone, is an interesting future work that we will like to follow up.
>
> We have added a citation to Girdhar et al. (2022) [5], and the discussion of this promising direction for future work in our revised version.
>
> **Modality gap in uni-modal setting**
> > If we consider a uni-modal setting, take Mean Teacher [1] in semi-supervised learning as an example: given an image, the consistency loss aims to minimize the predictions of a teacher and student model. The teacher model is the exponential moving average of the student model weights (i.e., the two models have different weights and operate on same data modality). Does the gap still exist here if we use a contrastive learning objective?
>
> Thank you for the question. Although there are two encoders in Mean Teacher [6], the weights of the two encoders are correlated in a very special way: namely, the teacher model is the exponential moving average of the student model weights [6]. This means that during training, the two cones produced by the two encoders are correlated in a special way beyond the contrastive learning objective. Our analyses assume that the two encoders are only connected via contrastive learning during optimization, and hence our analyses do not directly generalize to the Mean Teacher case [6].
>
> We have added a citation to Tarvainen et al. (2017) [6] and a discussion of this promising direction for future work in our revised version.

---

> > ### Author Response · Authors · 2022-08-02
> > **Response to Reviewer nReN (2/2)**
> >
> > **Modality Gap in unimodal contrastive learning**
> > > Is the proposed modality gap specific to multi-modal learning? Does the gap exist for unimodal contrastive learning?
> >
> > Thank you for the question. Unimodal contrastive learning typically only has one encoder as there is only one data modality, while Multimodal contrastive representation learning involves two separate encoders, which creates two different cones in the representation space. Our analyses assumed two separate encoders, and thus do not directly generalize to the one encoder case.
> >
> > **Simulating mismatched data**
> > > Section 4.3 (simulating mismatched data) seems an interesting investigation. What’s the multi-modal data being used here? Also, I wonder if the authors have some insight on how the modality gap correlates with multimodal data misalignment (e.g., an image is paired with a wrong caption due to data collection error).
> >
> > Thank you for the comments. The data used in Section 4.3 are synthetic 3-dim features for visualization purposes. This indicates that the presence of mismatched data might be an important forming factor of the modality gap under low temperatures. Regarding how multimodal data misalignment might contribute to the modality gap, this is a great question that we are interested in as well. We have added a sentence “Investigating how and to what extent the multimodal data misalignment could affect the contrastive loss landscape and thereby the modality gap is an interesting direction for future research.” in our revised version, and we think that answering this question precisely will require a separate publication with additional experimental and theoretical results.
> >
> >
> > **Improvements shown in Table 1**
> > > The improvement in Table 1 seems quite small. Is the reported number given by exhaustively iterating over all possible shifting? Are the results averaged over multiple runs / random seeds?
> >
> > Thank you for the question. We have shown that the improvements are statistically significant with p-value<0.01, and we refer the reviewer to the general response above. We have added the statistical testing to page 19 of the revised paper (See Supplementary Material).
> >
> >
> > We also clarify that we shifted embeddings along the line that passes through the two modality means. In other words, we did not exhaustively search for all possible directions. Because this embedding shifting procedure is deterministic (i.e., no training is involved),  our results are evaluated based on a single run.
> >
> >
> > We again thank Reviewer nReN for their review of our manuscript, and we hope that the above responses adequately address all concerns.

---

> ### Author Response · Authors · 2022-08-09
> **We would like to hear back from Reviewer nReN**
>
> Dear reviewer nReN,
>
> We would like to follow up to see if our response addresses your concerns or if you have any further questions. We would really appreciate the opportunity to discuss this further if our response has not already addressed your concerns. Thank you very much!

---

### Official Review · Reviewer_1atQ · 2022-07-11

**Rating:** 6
**Confidence:** 4
**Soundness:** 3 good
**Presentation:** 3 good
**Contribution:** 2 fair

**Summary:**

The paper investigates the geometric misalignment of multi-modal data representations in CLIP representation space. The authors investigate the existence of the gap under different random initialization, real or random input data and during the optimization of the model with contrastive learning objectives. They also provide theoretical analysis of the misalignment.


**Questions:**

I kindly ask the authors to elaborate on the following questions:
- In Sec 2.1, you investigate the cone effect on 3 pretrained models. I was wondering if this could be specific to the MSCOCO dataset and/or the selection of the 5000 embeddings?
- You never specify the dimension of the representations. How does this affect the gap? It is a known fact that higher dimensions lead to the diminishing effect of the distances.
- In general, I think it would be helpful to quantify the extent of misalignment. This could be done with geometric methods for evaluation of representations such as Delaunay Component Analysis (Poklukar et al, ICLR 20211),Improved Precision and Recall (Kynkäänniemi et al, NeurIPS 2019) or similar.
- The experiments suggest that the gap is present either when network parameters follow a Gaussian distribution (related to initialization) or the data (when using random noise). I assume the input data is also normalized before given to the network, and that the network uses batch or layer normalizations. Have you perhaps investigated whether the gap arises in more “non-Gaussian” setups?
Is the gap specific for the cosine similarity?


**Limitations:**

The experiments in the paper are specific to the CLIP model, which limits the scope of the work and makes it more difficult to judge the significance of the contribution. If no other models are added into the evaluation, I believe the introduction and abstract should reflect that, i.e., I do not think it is fair to claim that the work investigates the gap for general multi modal contrastive representation learning models if this is in fact not the case.

**Strengths And Weaknesses:**

Strengths:
The paper is well written and provides extensive analysis (both theoretical and empirical) of the geometric misalignment of representations under different modelling assumptions. Proofs and code are provided. The influence of the geometric misalignment is also investigated when using representations for real-world downstream tasks.

Weaknesses:
As far as I am aware, the observation about the geometric misalignment of multi-modal data representations is not novel but is typically referred to as heterogeneity gap (discussed for example, in https://ieeexplore.ieee.org/document/8715409).
While the experiments are interesting and insightful, I am doubtful about the significance of the results. In particular, I am not convinced by the results on the downstream tasks which are in my view the important ones. The authors investigate only one value of the modified gap and the results are not very significant (especially for Table 1).

---

> ### Author Response · Authors · 2022-08-02
> **Response to Reviewer 1atQ (1/2)**
>
> We thank Reviewer 1atQ for their positive comments and for providing thoughtful feedback on our work. We address many of Reviewer 1atQ’s comments in our general response above, and we provide additional details on specific comments below.
>
>
>
> **Adding Cone Effect Experiments on ImageNet**
> > the cone effect … could be specific to the MSCOCO dataset and/or the selection of the 5000 embeddings?
>
> We have added experiments on ImageNet to show that the cone effect is *not* specific to the MSCOCO dataset. We use the whole validation set of ImageNet, which contains 50,000 images, thereby scaling up the number of embeddings tested by an order of magnitude. We have added the ImageNet experiment results to page 21 of the revised paper (See Supplementary Material).
>
>
> | Dataset                   | ImageNet   | COCO       | ImageNet              | COCO                  |
> |---------------------------|------------|------------|-----------------------|-----------------------|
> |   Average cos similarity  |   ResNet   |   ResNet   |   Vision Transformer  |   Vision Transformer  |
> |   mean                    | **0.5160** | **0.5556** | **0.4835**            | **0.4679**            |
> |   std                     |   0.0618   |   0.0695   |   0.0883              |   0.0900              |
> |   25%                     |   0.4752   |   0.5081   |   0.4243              |   0.4095              |
> |   50%                     |   0.5142   |   0.5523   |   0.4778              |   0.4660              |
> |   75%                     |   0.5547   |   0.5993   |   0.5364              |   0.5222              |
>
> As shown in the table above, the average cos similarity and other statistics are similar on ImageNet and COCO, with only minor variations. In particular, the average cosine similarity on ImageNet is also substantially larger than 0, indicating that the embedding space is a narrow cone. This shows that the cone effect *not* specific to MSCOCO dataset and/or the selection of the 5000 embeddings.
>
>
>
>
> **Dimension of the representations**
> > You never specify the dimension of the representations. How does this affect the gap? It is a known fact that higher dimensions lead to the diminishing effect of the distances.
>
> The dimensions of the representations that we tested are: CLIP 512-dim, VideoCLIP 768-dim, ConVIRT 512-dim, and CLASP 768-dim. We have added this information in the revision (Page 16).
>
> We added an experiment to investigate how changing the embedding dimension of CLIP would affect the gap. We train 4 different multi-modal models from scratch using CLIP’s objective, with an embedding dimension of 64, 128, 256, and 512 respectively. We trained the models on Conceptual Captions 3M with 15 epochs. Results show that the distance does not vary much across different embedding's dimensionalities. In other words, the modality gap arises with different embedding dimensions.  We have added the experiment results to page 22 of the revised paper (See Supplementary Material).
>
>
> | Dim | Gap L2 Distance |
> |-----|-----------------|
> | 64  | 0.3545          |
> | 128 | 0.3440          |
> | 256 | 0.3377          |
> | 512 | 0.3512          |
>
>
> **The extent of misalignment (i.e.,modality gap)**
> > In general, I think it would be helpful to quantify the extent of misalignment. This could be done with geometric methods for evaluation of representations such as Delaunay Component Analysis (Poklukar et al, ICLR 2022), Improved Precision and Recall (Kynkäänniemi et al, NeurIPS 2019) or similar.
>
> Thank you for this suggestion. The extent of misalignment between CLIP's image embeddings and text embeddings (i.e., modality gap) is so large that both the precision and recall are zero for Kynkäänniemi et al [8]. More specifically, because CLIP's image embeddings and text embeddings are located in two *completely separate* regions of the embedding space, and they are perfectly linearly separable with a large margin (Supp. Figure 4), there would be *zero* shared support between the image embeddings and text embeddings, and thus both the precision and the recall are *zero*.
>
> We have added citations to Kynkäänniemi et. al. (2019) [8] and Poklukar et. al. (2022) [9], and a sentence in the discussion section “Development of geometric methods for evaluation of representations (Kynkäänniemi et. al. 2019, Poklukar et. al. 2022) to further capture the geometric landscape of the modality gap is also an interesting direction of future work.”

---

> > ### Author Response · Authors · 2022-08-02
> > **Response to Reviewer 1atQ (2/2)**
> >
> > **Adding Modality Gap Experiments in “non-Gaussian” setups**
> > > The experiments suggest that the gap is present either when network parameters follow a Gaussian distribution (related to initialization) or the data (when using random noise). I assume the input data is also normalized before given to the network, and that the network uses batch or layer normalizations.
> > > Have you perhaps investigated whether the gap arises in more “non-Gaussian” setups?
> >
> > Most of our results are in “non-Gaussian” setups, where i) the network parameters are pre-trained, and ii) the data are real images or texts. To make our settings even more “non-Gaussian”, we have added an experiment where we have iii) disabled both input data normalization (e.g., by ImageNet mean and std) and iv) all normalization layers.
> >
> >
> > New results are shown on page 22 of the revised paper (See Supplementary Material). The modality gap still clearly exists under such a “non-Gaussian” setup.
> >
> >
> >
> >
> > **Modality Gap is not specific to CLIP**
> > > The experiments in the paper are specific to the CLIP model, which limits the scope of the work and makes it more difficult to judge the significance of the contribution. If no other models are added into the evaluation, I believe the introduction and abstract should reflect that, i.e., I do not think it is fair to claim that the work investigates the gap for general multi modal contrastive representation learning models if this is in fact not the case.
> >
> > Thank you for the comments. The modality gap phenomenon we found is not limited to CLIP. As shown in Figure 1, we showed the modality gap phenomenon in not only CLIP, but also in various multi-modal contrastive representation learning models including VideoCLIP (videos + texts), ConVIRT (medical images + texts), and CLASP (amino-acid sequences + texts).
> >
> > We have also shown that our analyses are generalizable:
> > 1. We have shown that the cone effect is a general inductive bias of deep neural networks that hold on ResNet, vision transformer, and text transformer. We mathematically characterize the contraction mapping induced by linear layers with ReLU non-linearities to explain the cone effect, thereby confirming that the cone effect is a very general phenomenon.
> > 2. The contrastive learning objective we analyzed is also not limited to CLIP. This contrastive learning objective is one of the most widely adopted learning objectives in multi-modal contrastive representation learning models, which is used by VideoCLIP (videos + texts), ConVIRT (medical images + texts), CLASP (amino-acid sequences + texts), and many others.
> >
> > **Heterogeneity gap**
> > > the observation about the geometric misalignment of multi-modal data representations is not novel but is typically referred to as heterogeneity gap
> >
> > Thank you for the comments. We believe that the modality gap is a novel finding and is fundamentally different from the heterogeneity gap [4]. The heterogeneity gap states that the inherent differences in data modalities make exactly aligning the data representations from different data modalities (e.g., image, text) conceptually challenging for multimodal learning in general. However, this vague statement does not necessarily mean that the data representations from different modalities would be located in two **completely separate** regions of the embedding space, which is a much stronger statement. Therefore, the modality gap phenomenon is a novel finding, and also constitutes a much stronger statement than the heterogeneity gap.
> >
> > We have added a citation to Guo et al. (2019) [4] and a discussion to our revised version.
> >
> >
> > **Improvements shown in Table 1**
> > > I am doubtful about the significance of the results… the results are not very significant (especially for Table 1).
> >
> > Thank you for the comment. We have shown that the improvements are statistically significant with p-value<0.01, and we refer the reviewer to the general response above. We have added the statistical testing to page 19 of the revised paper (See Supplementary Material). We also reiterate that the main objective of our paper is to understand the modality gap phenomenon, a general inductive bias that holds across various data modalities and NN architectures. The goal of our paper is not to propose a method to close the gap or to improve downstream performance, which is an important direction of follow-up work.
> >
> >
> >
> >
> > **Thank you again for your feedback, which was very helpful in improving the paper.** We hope you would consider increasing your score in light of our detailed response. Please let us know if you have any more questions and we are happy to follow up!

---

> ### Author Response · Authors · 2022-08-09
> **We would like to hear back from Reviewer 1atQ**
>
> Dear reviewer 1atQ,
>
> We would like to follow up to see if our response addresses your concerns or if you have any further questions. We would really appreciate the opportunity to discuss this further if our response has not already addressed your concerns. Thank you very much!

---

### Official Review · Reviewer_2Ge4 · 2022-07-11

**Rating:** 4
**Confidence:** 4
**Soundness:** 2 fair
**Presentation:** 3 good
**Contribution:** 2 fair

**Summary:**

This manuscript demonstrates a modality gap phenomenon for multi-modal contrastive models like CLIP. Specifically, the authors analyze why the gap exists and its importance; (a) the modality gap is born at random initialization and the contrastive learning objective encourages the gap, and (b) changing the modality gap can affect zero-shot and fairness performances on downstream tasks. Furthermore, the authors provide a theoretical analysis of the modality gap phenomenon.

**Questions:**

I hope the authors could resolve my concerns in the weakness part above.

**Limitations:**

The authors do not provide any limitation or potential negative social impact of their work

**Strengths And Weaknesses:**

Strengths
- The writing is clear and easy to understand
- This manuscripts study interesting intrinsic phenomenon of contrastive-based multi-modal models (e.g., CLIP)
- Theoretical analysis supports the existence of the modality gap at randomly initialized weights

Weakness
Overall, the backups are not enough to support the claims and the improvements in the experiments are marginal
- The cone effect phenomenon in pretrained models; for example, the authors claim the avg 0.56 cosine similarity score (in Figure (a)) is high enough to show the embedding space is a narrow cone. But, compared to the random initialization (in Figure (b)), the score is already significantly reduced from 0.99 to 0.56. Is there any other baseline (such as standard contrastive learning on ImageNet) to show the values are meaningful?
- Inconsistent claims; the authors claim closing the gap increases the contrastive loss. But, Figure (d) contradicts the claim - closing the gap reduces the contrastive loss. In my understanding, the contrastive loss forces to close the gap between given pair of data (e.g., image and text pair), where the temperature parameter (i.e., sharpening parameter) in contrastive loss can control the magnitude of the loss. For example, Figure 8 in Supp. shows that high temperatures can remove the gap. So, after training, the gap could be negligible in cases.
- Experimental results are not supportive; improvements in Tables 1 and 2 seem to be marginal and simply obtained from the best results from the search space of hyperparameter $\lambda$ in Sec 4.2.
- The authors do not provide a proper guide for the modality gap; which one brings the benefits for multi-modal learning - closing or increasing the gap?

---

> ### Author Response · Authors · 2022-08-02
> **Response to Reviewer 2Ge4 (1/2)**
>
> We thank Reviewer 2Ge4 for reviewing our paper and providing helpful feedback on our work. We address many of Reviewer 2Ge4’s concerns in the general response above, and we provide additional details on specific comments below.
>
>
> **Cone effect: How narrow is the cone**
> > the authors claim the avg 0.56 cosine similarity score (in Figure (a)) is high enough to show the embedding space is a narrow cone. … Is there any other baseline (such as standard contrastive learning on ImageNet) to show the values are meaningful?
>
> Thank you for the comment. We clarify that the average 0.56 cosine similarity score already indicates that the embedding space is actually an *extremely narrow* cone in the 512-dimensional feature space. Cosine similarity ranges in [-1,1]. The following mathematical evidence can intuitively explain how narrow it is. We have added these discussions to page 3 of the revised paper (See supplementary material PDF):
> - **Fraction of surface area in a unit hypersphere:**
>   - In 2D, arccos(0.56)=55.94°, indicating that a cosine similarity of 0.56 can “occupy” 55.94°/360°=15.53% of the 2D unit circle.
>   - In 3D, cosine similarity of 0.56 can “occupy” $\frac{2 \pi r^2 (1- \cos \frac{55.94 \degree}{2})}{4 \pi r^2}$=3.34% of the 3D unit sphere.
>   - In 512D, cosine similarity of 0.56 can “occupy” less than $\frac{1}{2^{512}}$ fraction of the surface area in a unit 512D hypersphere.
>
> - **Gaussian baseline:** For random 512-dimensional vectors drawn from the standard normal distribution, the cosine similarity scores are zero-centered, with a standard deviation of 0.046. And note the at 0.56 cosine similarity score is much larger than 0±0.046.
>
>   - In fact, it is well known that any two high-dimensional vectors are likely to be orthogonal [7].
>   - To be more specific, for any fixed $y$ on a $n$-dimensional unit sphere and $\epsilon >0$, the following holds (Example 3.10 of Wainwright (2019)): $P( y^T Z > \epsilon/2 ) < \exp ^{-n \epsilon ^2 /8}$.
>   - This inequality implies that the cosine similarity between two random vectors on a $n$-dimensional unit sphere goes to zero with a high probability. With this result, the theoretical baseline can be set to zero, but our results in Figure 2-(a) show unintuitive results—even the smallest one, which is 0.56, is clearly greater than zero.
>
> Please let us know if you have any further questions regarding the narrowness of the cone effect and we are happy to follow up.
>
>
>  **Contrastive loss landscape and temperature**
> > Inconsistent claims; the authors claim closing the gap increases the contrastive loss. But…. In my understanding… the temperature parameter (i.e., sharpening parameter) in contrastive loss can control the magnitude of the loss.
>
> There is no inconsistency in our claims. In Line 187-195, we make two points about the contrastive loss landscape and the temperature:
> 1. **Under CLIP’s default temperature:** The default gap distance of 0.82 actually achieves the global minimum, (Figure 3(a), Line 187-190). Under CLIP’s default temperature, shifting toward closing the gap increases the contrastive loss.
> 2. **Higher than default temperature:** However, when the temperature increases, closing the gap becomes more optimal (Figure 3(c,d), Line 193-194). Note that CLIP’s default temperature is which is 1/100,  and Figure 3(c,d) uses much higher temperatures (e.g., 1/50, 1).
>
> There is no inconsistency between the two claims: claim (i) discussed the loss landscape under the fixed default temperature $\tau=1/100$, while claim (ii) discussed a temperature parameter that is higher-than-the default ones (e.g., 1/50, 1). We have clarified in the abstract that “During optimization, contrastive learning keeps the different modalities separated by a certain distance, which is influenced by the temperature parameter in the loss function.”

---

> > ### Author Response · Authors · 2022-08-02
> > **Response to Reviewer 2Ge4 (2/2)**
> >
> > **Improvements shown in Table 1 and 2**
> > > improvements in Tables 1 and 2 seem to be marginal
> >
> > Thank you for the comment. We have shown that the improvements are statistically significant with P-value<0.01, and we refer the reviewer to the general response above. We have added the statistical testing to page 19 of the revised paper (See Supplementary Material). We also reiterate that the main objective of our paper is to understand the modality gap phenomenon, a general inductive bias that holds across various data modalities and NN architectures. The goal of our paper is not to propose a method to close the gap or to improve downstream performance, which is an important direction of follow-up work.
> >
> >
> > **How should we modify the modality gap**
> > > The authors do not provide a proper guide for the modality gap; which one brings the benefits for multi-modal learning - closing or increasing the gap?
> >
> > Thank you for the comment. We agree this is an important question, but the goal of our paper is mainly to analyze how the modality gap is formed. This is much more complex than simple intuition, where we find that it is related to many fundamental questions in machine learning, such as initialization and optimization. Systematic analysis of the impact of the gap on applications is an important direction for future work.
> >
> > In terms of the future impact of this line of research, it has been shown that intervening the modality gap can improve the performance of both image-to-text retrieval on MS COCO and Flickr30k [1], and personalized image retrieval [2]. Closing the modality gap can also simplify DALLE-2 by removing the prior network which converts text embedding to image embedding [3]. We believe this would be a promising direction for future work.
> >
> >
> > **We again thank Reviewer 2Ge4 for their review of our manuscript. Your questions have improved the paper.**  We hope you would consider improving your score in light of our detailed response. Please let us know if you have any more questions and we are very happy to follow up.

---

> ### Author Response · Authors · 2022-08-05
> **We would like to hear back from Reviewer 2Ge4**
>
> Dear reviewer 2Ge4,
>
> We would like to follow up to see if our response addresses your concerns or if you have any further questions. We would really appreciate the opportunity to discuss this further if our response has not already addressed your concerns. Thank you very much!

---

> > ### Comment · Reviewer_2Ge4 · 2022-08-06
> > **Response to the authors**
> >
> > Thanks for the authors' comments. However, there are still parts I am not convinced of.
> > - About the contrastive loss and its temperature:  If your claim "Contrastive learning preserves modality gap" does not hold for higher temperatures, as shown in Figure 8, it sounds overclaimed. Specifically, Figure 8 shows that fine-tuning CLIP with the contrastive loss under highly changed temperatures (t=1/10 and 1) does not preserve the gaps anymore. Overall, I still think the gaps in pre-trained models are not from the contrastive loss, just surmountable issues from their training recipes.
> > - Why studying the gap is important: As the authors pointed out in the discussion section (``...why studying the gap is important, i.e., it can affect the downstream task performance and fairness...``), it would be one of the potential readers' concerns to understand the importance of the gap. Although the authors remarked that ``...The goal of our paper is not to propose a method to close the gap or to improve downstream performance...`` in the general response, I expect to hear another explanation if proposing a method to close the gap or improving downstream performance was not one of the main contributions.
> >
> > I hope the authors can add more detailed discussions for the above.

---

> > > ### Author Response · Authors · 2022-08-08
> > > **Follow-up Response to Reviewer 2Ge4 (1/2)**
> > >
> > > Thank you very much for your comments! We really appreciate your time.
> > >
> > > **About the contrastive loss and its temperature**
> > > > If your claim "Contrastive learning preserves modality gap" does not hold for higher temperatures, as shown in Figure 8, it sounds overclaimed. Specifically, Figure 8 shows that fine-tuning CLIP with the contrastive loss under highly changed temperatures (t=1/10 and 1) does not preserve the gaps anymore.
> > >
> > > Thank you for this question. We clarify that there is still a gap with the higher temperature during fine-tuning in Figure 8 right part. Fine-tuning with temperature=1 still leads to a *significant gap with a distance of 0.24*. Moreover, in Figure 8 left part, we clarify that while there seems to be no gap in the PCA view for temperature=1, there is a gap in the *high dimensional space*. We can think of a simple example where two spheres with a radius of 100 are in z=0.1 and z=-0.1. If we use PCA to reduce to 2D, there will be no gap from the 2D PCA view but the actual gap=0.2 in 3D. Therefore, our finding that "contrastive learning preserves modality gap" *does* hold for higher temperatures. Simply changing temperatures cannot eliminate the gap.
> > >
> > >
> > > **The bigger picture: Why studying the gap is important**
> > > > I hope the authors can add more detailed discussions for the above.
> > >
> > > Thank you for the suggestion. There has been tremendous recent interest and excitement in studying the **inductive bias of neural networks** mathematically and empirically [10-21]. For example, an influential line of research shows that neural networks can easily fit random labels [10], and SGD provides an inductive bias of “implicit regularization” by favoring minima that are flatter [11] and closer to the initialization [12]. Another impactful line of research shows that neural networks trained on natural scenes are biased towards texture [13], and exhibit gestalt closure similar to human perception, which is an inductive bias long-studied in the psychology literature [14]. Researchers have also shown that neural networks favor “shortcut learning”, which may be a common characteristic of learning systems, biological and artificial alike, as known in Comparative Psychology, Education and Linguistics [15,16].
> > > Our paper contributes to this broad and exciting trend of studying the inductive bias of neural networks by analyzing the modality gap phenomenon which occurs consistently in multi-modal contrastive representation learning.
> > >
> > > By studying the modality gap, our analyses also provide new insights into the cone effect, which we show is a general inductive bias of deep neural networks. In the recent literature, the cone effect has been observed in the language representations from language models such as BERT and GPT-2 [19,20,21]. A common explanation is that the *unbalanced* distribution of word frequencies biased the optimization [20,21]. However, we found that the cone effect still exists in models with random weights  (Figure 2(c)). In fact, the average cosine similarity there is even higher than in trained models. For example, any two embeddings from a randomly initialized ResNet have on average an almost perfect (0.99) cosine similarity. Interestingly, the cone effect still holds when the input data is random noise, indicating that the unbalanced data distribution suggested in previous works is not necessary for the cone effect. Together these experiments suggest that the cone effect reflects a more general inductive bias of deep networks than might be previously appreciated. We rigorously analyzed why it happens in Theorem 1 and further examined how the cone effect leads to the modality gap when there are multi-modality data (Theorem 2 and Figures 2 and 3).
> > >
> > > To sum up, there has been a long-established line of influential research in studying the **inductive bias of neural networks** mathematically and empirically including the cone effect, and our research makes novel contributions to and pushes the boundaries of knowledge significantly forward in this impactful research topic for multimodal models.
> > >
> > > **Please let us know if you have further questions and we are happy to further respond!** If our responses (this and the previous one) have addressed some of your questions, we would very much appreciate it if you would consider increasing your score.

---

> > > > ### Author Response · Authors · 2022-08-08
> > > > **Follow-up Response to Reviewer 2Ge4 (2/2)**
> > > >
> > > > **References**
> > > >
> > > > [10] C. Zhang, S. Bengio, M. Hardt, B. Recht, and O. Vinyals. Understanding deep learning (still) requires rethinking generalization. Commun. ACM, 64(3):107–115, 2021.
> > > >
> > > > [11] N. S. Keskar, D. Mudigere, J. Nocedal, M. Smelyanskiy, and P. T. P. Tang. On large-batch training for deep learning: Generalization gap and sharp minima. In ICLR, 2017.
> > > >
> > > > [12] B. Neyshabur, Z. Li, S. Bhojanapalli, Y. LeCun, and N. Srebro. The role of over-parameterization in generalization of neural networks. In ICLR, 2019.
> > > >
> > > > [13] R. Geirhos, P. Rubisch, C. Michaelis, M. Bethge, F. A. Wichmann, and W. Brendel. Imagenet-trained CNNs are biased towards texture; increasing shape bias improves accuracy and robustness. In ICLR, 2019.
> > > >
> > > > [14] B. Kim, E. Reif, M. Wattenberg, S. Bengio, and M. C. Mozer. Neural networks trained on natural scenes exhibit gestalt closure. Computational Brain & Behavior, 4(3):251–263, 2021.
> > > >
> > > > [15] R. Geirhos, J. Jacobsen, C. Michaelis, R. S. Zemel, W. Brendel, M. Bethge, and F. A. Wichmann. Shortcut learning in deep neural networks. Nat. Mach. Intell., 2(11):665–673, 2020
> > > >
> > > > [16] D. Arpit, S. Jastrzebski, N. Ballas, D. Krueger, E. Bengio, M. S. Kanwal, T. Maharaj, A. Fischer, A. C. Courville, Y. Bengio, and S. Lacoste-Julien. A closer look at memorization in deep networks. In ICML, 2017.
> > > >
> > > > [17] J. Frankle and M. Carbin. The lottery ticket hypothesis: Finding sparse, trainable neural networks. In ICLR, 2019.
> > > >
> > > > [18] Z. Allen-Zhu, Y. Li, and Z. Song. A convergence theory for deep learning via over-parameterization. In ICML, 2019.
> > > >
> > > > [19] K. Ethayarajh. How contextual are contextualized word representations? comparing the geometry of BERT, ELMo, and GPT-2 embeddings. In EMNLP, 2019.
> > > >
> > > > [20] J. Gao, D. He, X. Tan, T. Qin, L. Wang, and T. Liu. Representation degeneration problem in training natural language generation models. In ICLR, 2019.
> > > >
> > > > [21] B. Li, H. Zhou, J. He, M. Wang, Y. Yang, and L. Li. On the sentence embeddings from pre-trained language models. In EMNLP, 2020.

---

### Author Response · Authors · 2022-08-02
**Response to all reviewers (1/2)**

We thank the reviewers for their thoughtful and constructive review of our manuscript. We were encouraged to hear that the reviewers found the modality gap phenomenon we present to be interesting (2Ge4, nReN, QxpT), original (QxpT), and insightful (nReN), and that they view our analysis as extensive (nReN, 1atQ), solid and well-supported (QxpT), and that all reviewers found our paper well-written and clearly organized (2Ge4,1atQ, nReN, QxpT). We have carefully updated the paper (*PDF uploaded as Supplementary Material*) based on the reviewers’ suggestions. In response to feedback, we provide a general response here to points raised by multiple reviewers, and individual responses below to address each reviewer’s concerns.


In response to the general comments about **the main objective and the contributions of our paper**, we reiterate that the main objective of our paper is to understand the modality gap phenomenon, a general inductive bias that holds across various data modalities and NN architectures. The goal of our paper is not to propose a method to close the gap or to improve downstream performance, which is an important direction of follow-up work. In summary, our paper makes the following contributions:
1. **Demonstrating a general modality gap phenomenon**: To the best of our knowledge, we demonstrate a general modality gap phenomenon for the first time. We show that this phenomenon holds across a large class of networks and multi-modal problems and hence is likely to be broadly applicable to the entire field of multi-model learning. In the revision, we added more experiments supporting our findings using the ImageNet dataset.
2. **Explaining why the gap occurs**: To explain the modality gap, we provide a three-part explanation supported by extensive theoretical and empirical analyses.
    - **Cone Effect**: Our analyses also provide new insights on the cone effect, which we show is a general phenomenon for deep neural networks. Our findings and analyses on the cone effect contradict previously held notions and advance scientific understanding.
3. **Theoretical analyses**: We mathematically characterize the contraction mapping induced by linear layers with ReLU non-linearities to explain the cone effect. Our theory matches well with experiments and provides insights for understanding the general inductive biases of deep neural networks.

Regarding **the improvements shown in Table 1**: We thank the reviewers for asking about this. In this revision, we have further investigated the significance of the improvements in Table 1, showing that they are *statistically significant*. Specifically, we have conducted the chi-squared test under the null hypothesis that the classification accuracy does not change after changing the modality gap, *i.e.*, $H_0 : p_{\mathrm{before}} =  p_{\mathrm{after}}$. Our results show that the p-values are less than $0.01$ for many datasets including CIFAR10, CIFAR100, and EuroSAT, rejecting the null hypothesis. We use the whole dataset instead of only the validation set to make our results more robust because our embedding shifting experiments involve no fine-tuning. We have added the statistical testing to page 19 of the revised paper.


| Dataset | Original Acc | Modified Acc | Direction |  p-value  |
|----------|:--------:|:--------:|:---------:|:---------:|
| CIFAR10  | 0.9026   | 0.9104   | ↑         | 3.476e-06 |
| CIFAR100 | 0.6705   | 0.6776   | ↓         | 8.701e-03 |
| EuroSAT  | 0.5494   | 0.5686   | ↓         | 7.020e-06 |



We would again like to thank all reviewers for their time and feedback, and we hope that our changes adequately address all concerns.

---

> ### Author Response · Authors · 2022-08-02
> **Response to all reviewers (2/2)**
>
> **References**
>
> [1] So, Junhyuk, Chang-Seok Oh, Minchul Shin and Kyungwoo Song. Multi-Modal Mixup for Robust Fine-tuning. arXiv:2203.03897 [cs.CV], Mar 2022
>
> [2] Cohen, Niv, Rinon Gal, Eli A. Meirom, Gal Chechik and Yuval Atzmon. This is my unicorn, Fluffy: Personalizing frozen vision-language representations. arXiv:2204.01694 [cs.CV], April 2022.
>
> [3] Ramesh, Aditya, Prafulla Dhariwal, Alex Nichol, Casey Chu, and Mark Chen. "Hierarchical Text-Conditional Image Generation with CLIP Latents." arXiv:2204.06125 [cs.CV], April 2022.
>
> [4] W. Guo, J. Wang and S. Wang, "Deep Multimodal Representation Learning: A Survey," in IEEE Access, vol. 7, pp. 63373-63394, 2019, doi: 10.1109/ACCESS.2019.2916887.
>
> [5] Girdhar, Rohit, et al. "Omnivore: A single model for many visual modalities." CVPR (2022).
>
> [6] Tarvainen, Antti, and Harri Valpola. "Mean teachers are better role models: Weight-averaged consistency targets improve semi-supervised deep learning results." NIPS (2017).
>
> [7] Wainwright, Martin J. High-dimensional statistics: A non-asymptotic viewpoint. Vol. 48. Cambridge University Press, 2019.
>
> [8] Poklukar, Petra, et al. "Delaunay component analysis for evaluation of data representations." ICLR (2022).
>
> [9] Kynkäänniemi, Tuomas, et al. "Improved precision and recall metric for assessing generative models." NeurIPS (2019).

---

### Meta-Review · Area_Chair_J4BB · 2022-08-30

**Recommendation:** Accept
**Confidence:** Less certain

**Metareview:**

This paper investigates the gap between representations when training with a contrastive objective, through the characterisation of the gap in various settings, and building a theoretical analysis of this gap.

The reviewers mostly agree that the paper tackles an interesting problem through investigation and characterisation of the inductive biases provided by CLIP-style models, and the experiments appear to cover a good number of cases.

The primary issues with the work however appear to be with some of the framing---it comes across as an investigation into something a bit more generic than the title suggests, and the claims to novelty, while reasonable, are also a bit too strong given the existence of the heterogeneity gap. The authors argue that finding that multi-modal data project to separate subspaces is somewhat reasonable, but I still don't think that supports as strong a claim as given.

On balance, though it appears as if the paper has more merits than issues, and most of the issues raised could be addressed with a bit of work. I would strongly urge the authors to actually make the edits for framing, clarity, and incorporating the additional experiments from the rebuttal into the manuscript, as requested by the reviewers.

**Award:**

No

---

### Decision · Program_Chairs · 2022-09-14

Accept